# High-throughput methods leveraging robotics and computer vision for the development of therapeutic phage cocktails

Taylor J. R. Penke [1] ✉, Aeron Tynes Hammack [1,2] ✉, Lana J. McMillan[1], Ethan Baker[1], Pearl Wilcock[1], Nick Healy[1,3], Morgan K. Y. Wall[1,2], Naomi Chavez[1], Iain Wright[1], Hannah H. Tuson [1], Sara Woessner[1], Ashley Trama[1], Cameron J. Prybol [1], Eyra Dordi[1], Ava Ghobadian[1], David G. Ousterout[1], Nicholas R. Conley[1] & Paul Garofolo[1]

We present the high-throughput automated screening techniques that are being used to develop bacteriophage-based therapeutic products currently under investigation in human clinical trials to combat urinary tract infections[1]. By integrating modern liquid handling robotics, standardized phenotypic assays, and computer vision-based enumeration, we established a platform capable of reproducibly screening large collections of phages against clinically derived bacterial strain panels. This approach enabled systematic assessment of phage-bacteria interactions at scale, facilitating the identification and optimization of phage cocktails with broad in vitro activity. Although bacteriophage therapy has long been investigated as a strategy for treating bacterial infections, few frameworks exist for developing phage combinations in a reproducible and scalable manner. The methods outlined here address this gap and aim to support the broader development of therapeutic assets available to combat antibiotic resistance.

The ability of bacteria to evolve and acquire resistance to antibiotics poses a significant threat to global public health[2–5]. In 2019, an estimated 1.27 million deaths were attributed to bacterial antimicrobial resistance (AMR)[2], and deaths associated with AMR are estimated to reach 8.22 (6.85–9.65) million by 2050[4,5]. *Escherichia coli* (E. coli) was associated with the most bacterial AMR deaths—approximately 800,000—in 2019[2]. Although broad-spectrum antibiotics are crucial tools for both preventing and treating a variety of infections, over-prescription has contributed to increasing rates of AMR[6,7]. Additionally, their non-specific nature may disrupt the natural balance of patient microbiomes, leading to a dysbiosis that allows opportunistic pathogens to fill niches opened by antibiotic treatment[8]. Despite efforts in place to prevent unnecessary antibiotic use[9], alternative therapeutics are needed to provide specific

and effective treatment while mitigating the rise of infections caused by resistant pathogens.

Urinary tract infections (UTIs) are a leading indication for antibiotic usage, representing between 1–6% of all medical visits[10], and impacting over 400 million people annually[11]. Uncomplicated UTIs are disproportionately common in women, with a lifetime incidence of 50–60%[10,12]. Although many UTIs are relatively easy to treat with traditional antibiotics, recurrent UTIs are common, with 30–50% of women experiencing a repeat infection[13]. Uropathogenic E. coli (UPEC) is the primary causative agent in UTIs, with a prevalence of 75–80% in uncomplicated UTIs and 65% in complicated UTIs[10,14].

Over the last decade, bacteriophage (phage) therapy has garnered increasing interest as an alternative treatment modality. Phages are ubiquitous viruses that infect bacteria and are the most

[1]Research and Development, Locus Biosciences, 523 Davis Drive, Morrisville, North Carolina, US. [2]Molecular Foundry, Lawrence Berkeley National Laboratory, 1 Cyclotron Road, Berkeley, California, US. [3]Automation Research, Hamilton Company, 4970 Energy Way, Reno, Nevada, US. ✉e-mail: taylor.penke@locus-bio.com; athammack@lbl.gov

abundant biological entity on Earth. Phages also possess an incredible spectrum of genetic diversity, with different phages exhibiting unique and highly selective mechanisms of action, giving them a subspecies level specificity for different bacterial strains. This specificity and their favorable safety profile[15] make phages an attractive therapeutic target. However, the vast quantity and heterogeneity of phages pose both a significant opportunity and a challenge in discovering the appropriate phages and their optimal combination for therapeutic development. Phage therapy can be applied via personalized or fixed cocktails. Personalized phage cocktails are tailored to each individual, offering the advantage of targeted and specific development[16,17]. In contrast, fixed phage cocktails are designed for broad use across an indication, and although they require more extensive upfront development, ultimately can be deployed to more patients at lower costs.

In this work, our goal was to establish repeatable and automated unit operations that allow fixed phage cocktail development with efficacy across pathogenic strain panels. Here, we report an automated platform designed for high-throughput discovery and activity assessment of phages and phage cocktails. This platform has enabled the standardized assessment of more than 3.8 million phage and bacteria reactions. As an example of its utility, we describe the methods used to develop a phage cocktail targeting E. coli isolates from UTIs. The candidate cocktail demonstrates in vitro activity against 96.4% of 356 uropathogenic E. coli (UPEC) isolates and reduces bacterial burden by ≥99.999% in 94% of those isolates. Among 38 E. coli isolates obtained from participants in the Part 1 open-label portion of a Phase 2b trial, 97% (37/38) exhibit in vitro sensitivity to the cocktail.

## Results

### Defining a clinically relevant strain panel to represent E. coli urinary tract infection diversity

We first set out to define a target panel of strains to represent the epidemiologically relevant diversity of UPEC strains. We acquired 1773 strains of E. coli from commercial and academic sources, as well as our own clinical trials, including a Phase 0 surveillance study and a Phase 1b safety study. Approximately 10,000 E. coli strain genomes were acquired from the BV-BRC (formerly PATRIC) database[18], and their genomic diversity was compared to genomes acquired from our E. coli strain bank (Supplementary Fig. 1a). Whole genome comparisons indicated that our bank contained representatives of almost all genomic clusters. Follow-on analyses indicated strong representation of sequencing types found in a meta-analysis of extraintestinal pathogenic E. coli strains[19]. Representatives of the top 20 sequencing types, which composed 85% of E. coli sequencing types found across 217 studies, were included in our collection (Supplementary Fig. 1b). In addition to genomic content, we also considered a variety of phenotypic and clinical attributes of strains, including patient gender, antibiotic resistance profile, geography of isolation site, and year of isolation.

Of the available strains, we selected 356 to establish a UTI Clinical Panel (Fig. 1). All strains were isolated from female urinary tracts and included isolates from patients living in 39 US states (see "Methods"). In total, 98% of isolates were collected between 2019 and 2021 (contemporaneous at the time of panel creation), and 29% were multidrug-resistant. To ensure repeatable growth of bacteria across assays, liquid handlers were used to normalize each strain titer via optical density (OD), and multiple aliquots from the same initial culture were generated and frozen (see "Methods"). Samples were added to pre-barcoded, automation-compatible tubes to maintain sample traceability and limit cross-sample contamination, which were then stored in automated biobanks (see "Methods"). This panel served as the source of strains for subsequent phage discovery and cocktail optimization.

### High-throughput robotics design for engineered phage cocktail development

A major challenge in developing a phage cocktail for use across a patient population is identifying the optimal combination of phages with an ability to infect diverse pathotypes and prevent resistance development for the full range of pathogenic E. coli. As collections of phages increase, testing all possible combinations quickly becomes unfeasible with current technologies (Fig. 1b). For example, testing all cocktails composed of 6 phages from a 500-phage collection would require creating $2.1 \times 10^{13}$ cocktails. The situation becomes even less tractable when one considers the number of replicates required to achieve statistical power and all of the infection-condition variables—including bacterial strains, growth media, phage input titer, and bacterial input titer—for which the antibacterial activity of each cocktail can be assessed.

To address this challenge, we designed a high-throughput robotic platform capable of monitoring phage and bacteria interactions across a range of phenotypic assays. We reasoned that a sufficiently sized database of interactions would enable machine learning approaches to predict cocktail efficacy of untested combinations. To this end, we designed two VANTAGE LIQUID HANDLING SYSTEM® work cells in collaboration with Hamilton Company (Fig. 1c). Each work cell contains a pipetting module consisting of 8 spanning pipette channels and a fixed 96-channel pipettor for flexible liquid handling. The base of the work cell contains a plate sealer, waste storage, and an Entry/Exit module allowing high-volume labware retrieval via built-in elevators. In the rear cabinet, dual shaking incubators provide 88-plate capacity and two plate readers allow wavelength readings (Fig. 1d). A sliding robotic arm between the front and rear cabinets is used to transfer labware throughout the work cell. In total, each system is capable of high-throughput execution of most microbiological assays.

### Phage discovery and phenotypic characterization

We began phage discovery efforts by establishing relationships with wastewater facilities throughout the United States. We acquired 1084 environmental samples from 577 providers in 48 states. These samples were arrayed across individual bacteria strains from the Clinical Panel, incubated with shaking at 250 rpm to allow phage amplification, and plaqued via the double agar overlay method on indicator strains. Phages were isolated via triple plaque purification and amplified in liquid culture. E. coli targeting phages ($n = 1143$) were banked, and 421 were selected based on their titer post-amplification. Each phage was sequenced, and its genome was screened for virulence genes, including antibiotic resistance genes, and used to predict phage lifestyle (i.e., temperate or obligately lytic).

To assess phage host range, we developed an automated optical-density-based assay on our work cells capable of testing 15,000 reactions per instrument per run (see "Methods"). In this method, the pipetting module adds various combinations of phages, bacterial strains, and media to 384-well plates accessed from the elevator system. Completed plates are moved into the shaking incubators for bacterial growth and transferred to plate readers for optical-density time-course readings routinely over a 20 h period. Growth curves from each well containing phages and bacterial strains were compared to a control well containing only bacteria via area-under-the-curve (AUC) measurements (Fig. 2a, see "Methods"). All analyses were performed by version-controlled Continuous Integration/Continuous Deployment (CI/CD) software pipelines that store run metadata and results in a centralized cloud database ("Methods").

We began assessment of individual phage efficacy by testing the collection of 421 phages against stochastically chosen 44- or 88-strain subsets of our full Clinical Panel. Initially, the bacteria susceptibility profile for each phage was examined, and strains with limited phage susceptibility were prioritized for subsequent waves of phage discovery. Additionally, phages were assessed against bacteria

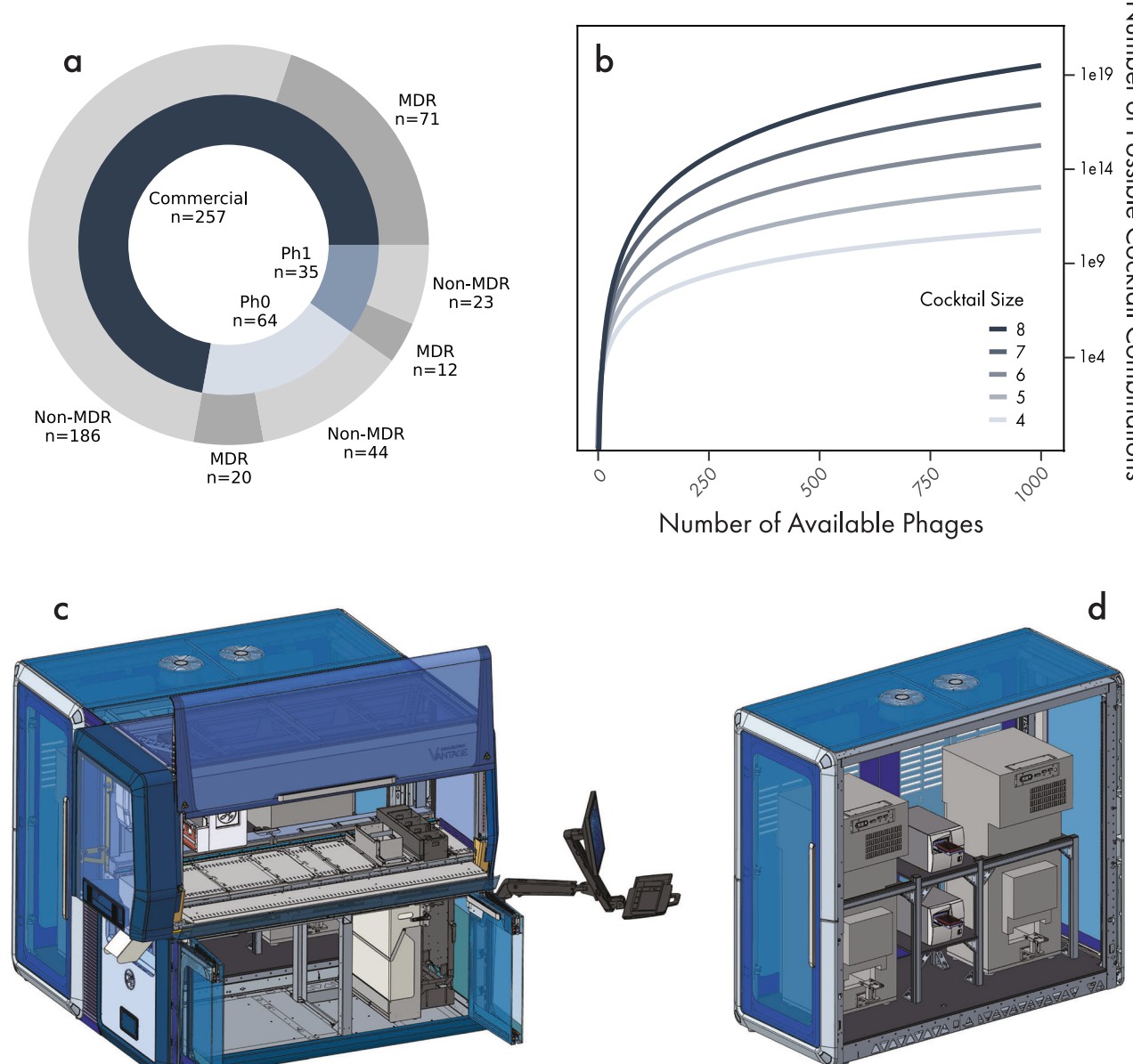

**Fig. 1 | High-throughput robotics design to enable development of an engineered phage cocktail targeting *E. coli*. a** A panel of 356 E. coli strains isolated from female patients with urinary tract infections was selected from a bank of approximately 1700 isolates. The inner circle shows the number of panel strains purchased from commercial sources or acquired during our Phase 0 (Ph0) surveillance study or Phase 1 (Ph1) trial, unpublished datasets. The outer circle indicates the number of multi-drug-resistant (MDR) isolates defined as resistant to 3 or more antibiotic classes. **b** Number of possible cocktail combinations with a variable number of available phages and the number of phages in the cocktail. **c** Render of Hamilton VANTAGE system designed for testing phage and bacteria interactions. The top portion of the unit houses a gantry system with 8 spanning channel pipettors and a fixed 96-channel pipetting unit. The bottom portion of the unit houses an elevator to provide continual access to plates and tips, along with waste disposal and a plate sealer. **d** Render of the rear portion of the Hamilton VANTAGE system housing two shaking Liconic incubators and two Biotek Powerwave HT plate readers. Panels c and d are reproduced with permission from Hamilton Company. © 2025 Hamilton Company. All rights reserved.

grown in different matrices and at varying phage and bacteria input titers to better understand the consistency of phage activity in different environments.

We next set out to identify the minimum effective combination of phages that would have strong antibacterial activity and durable suppression of all UTI isolates in our panel. Because all cocktail combinations could not be tested, we developed a proprietary cocktail prediction model that leveraged our database of phage and bacteria reactions to recommend cocktails for additional testing. Outcomes from the tested cocktails were then reincorporated into the prediction engine. In total, we collected 1.5 million growth curves (Fig. 2c)

assessing the efficacy of phages and phage cocktails. AUC ratio distributions were bimodal with peaks centered around 0.1 (the lowest optical density reading with media in the optical path) and 1 (no difference between phage-treated and control growth curves). The AUC ratio value of 0.65 was approximately equidistant from the two peaks and was used as a threshold value for classifying suitable phage activity. Using these data, we narrowed to 28 top-performing cocktails and tested their activity against the full Clinical Panel at a titer of $1 \times 10^8$ Plaque Forming Units per mL (PFU/mL). These cocktails possessed host ranges between 89–95% and exhibited similar bacteria susceptibility profiles (Fig. 2d). It is worth noting that the measured host range

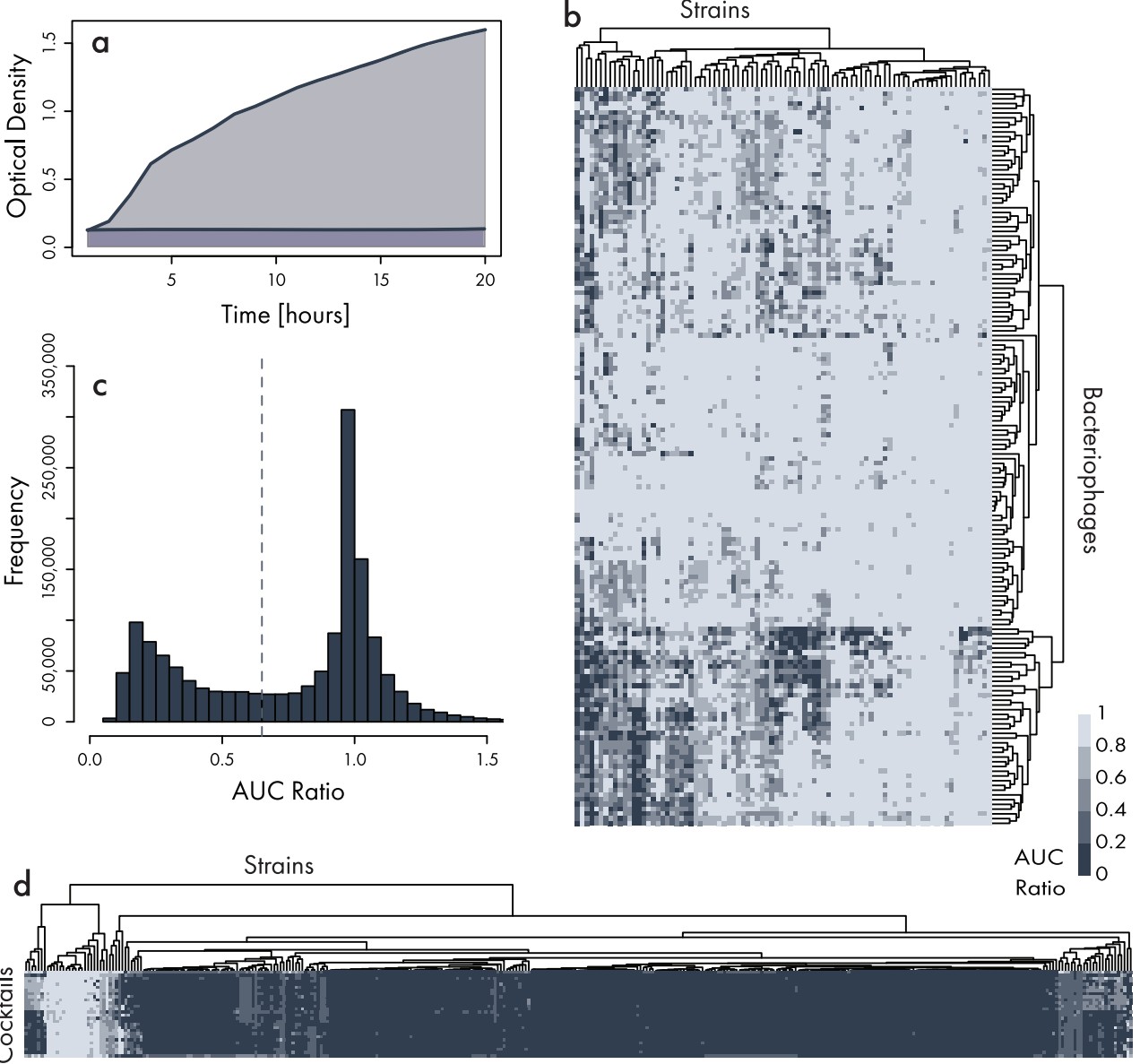

**Fig. 2 | Automated screening of E. coli phage bank. a** Example growth curves established from optical-density time-course readings across a 20-h time frame for bacteria-only controls and bacteria treated with phage. Area Under the Curves (AUC) represented by gray shading (bacteria only) and blue shading (bacteria and phage) are calculated, and an AUC ratio is determined by dividing the phage-treated AUC by the control AUC, such that lower values indicate greater growth suppression (i.e., phage efficacy). **b** Representative heatmap showing a subsample of individual phages tested against a panel of 88 bacterial strains. Rows and columns are hierarchically clustered using Euclidean distances. **c** Histogram showing the distribution of 1.5 million AUC ratios collected to characterize bacteria and phage interactions. The gray dotted line indicates the approximate center of the bimodal distribution and is used as a threshold for phage activity classification. **d** Heatmap showing efficacy of the top-performing cocktails against the full clinical panel of UTI isolates ($n = 356$). Phage-bacteria combinations that produce lower AUC ratios, indicative of better efficacy, are shown in darker colors.

activity for the top-performing cocktails exceeds the expected host ranges that simple algorithmic cocktail selection by random choice or by simply selecting the top six phages with the highest host ranges would predict, as shown in Fig. 3.

**Automated colony enumeration via computer vision**

Because cocktail activity was similar between many of our top cocktails, we tested cocktail activity in complementary antibacterial assays to identify the best cocktail. A significant limitation of the liquid culture optical-density-based assay is a high lower limit of detection, which ranges between $10^7$ and $10^8$ Colony Forming Units per mL (CFU/mL). Alternative quantification approaches include qPCR and serial dilution, and spotting. We selected serial dilution and spotting to

measure viable bacteria in a reaction rather than the measurement of DNA, which does not report on bacteria viability.

In this assay, phages, strains, and media were added to 96-well plates via liquid handlers and transferred to shaking incubators (see "Methods"). At varying time points, plates were extracted from the incubator, serially diluted across 7 additional microtiter plates using the work cell's 96-channel pipettor (Fig. 4a, Supplementary Fig. 3), and $2\,\mu L$ from each well was spotted onto barcoded agar plates to allow colony formation (Fig. 4b). Imaged plates were then subjected to a custom image analysis pipeline, written in Python, to enumerate the colony and plaque forming units from the images of the microtiter plates and to determine the original sample titer (Fig. 4c–e, "Methods"). The computer vision components of this pipeline leveraged

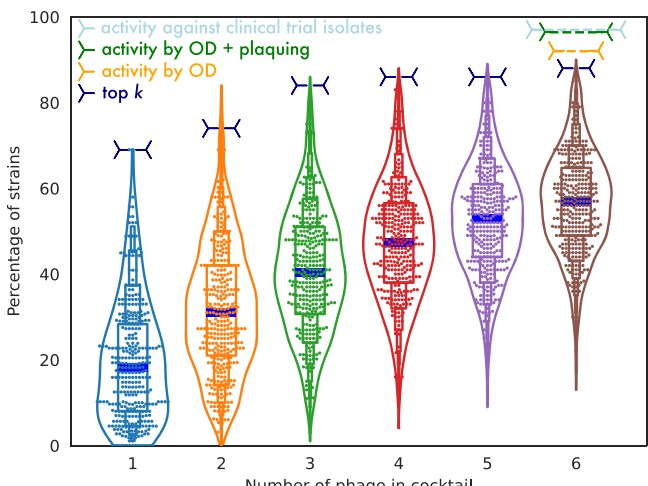

**Fig. 3 | Comparison of host range for simple cocktail selection methods to final chosen cocktail host range.** This figure shows the projected host ranges that would be expected for two naive cocktail selection methods, "random choice of $k$ phage(s)", and "top $k$ performing phage(s)". The inputs to the simple cocktail selection host range simulations were the individual host range performances from a subset of $N = 170$ E. coli phages. The simulated host range is calculated by summing the host range hit-versus-no-hit vectors without consideration of potential synergistic or antagonistic interactions between individual phages. The swarm and violin plots show the simulated host range results for sampling at random to form possible cocktails. The swarm plots reflect sampling 340 random putative cocktails for all numbers of phage in the simulated cocktails. These swarms were sampled from simulated cocktails reflecting the full number of possible cocktails for $k = 1, 2$, amounting to $\binom{N}{k} = \binom{170}{1} = 170$, $\binom{170}{2} = 14,365$. For $k > 2$, only 14,365 simulated cocktails were sampled for the swarm plots, given the large number of possible phage combinations. The navy blue lines indicate the results of combining $k = 1, 2, 3, 4, 5, 6$ different phages, selected in order of descending host range, into simulated cocktails combining the "top $k$ performers". The orange dashed line shows the 92% in vitro liquid OD host range activity of LBP-EC01 against the UPEC strain panel, and the dark green dashed line shows the 96.4% in vitro activity of LBP-EC01 against the UPEC strain panel when the plaquing host range is added to complement the liquid OD host range. The light blue dashed line shows the 97% in vitro activity of LBP-EC01 against the 38 E. coli isolates obtained from clinical trials. The enhanced performance of LBP-EC01 relative to the predicted wild-type host range cocktail candidates likely reflects the increased efficacy of the engineered phages that are included in the final cocktail.

three distinct algorithms to provide internal checks and balances to determine whether a human investigation of the final titers would be merited. The three algorithms employed were: 1) Watershed = basin finding and watershed marking, 2) Integrated = integrated optical density, and 3) Circle = feature detection by Hough circle transform. The sample enumeration was conducted on a high-performance cloud platform, managed by Apache Airflow for task scheduling and Dask for task-specific parallelism. Configuration parameters were managed through GitHub, triggering Google Cloud Build to execute tasks in precompiled containers, ensuring reproducibility and clear data provenance with automated cloud backups (see "Methods").

Across all experiments, over 2.2 million spots have been automatically enumerated using this method. Figure 5 shows the aggregated $\log_{10}$ titering results for approximately 500,000 CFU measurements across 85 unique experiments. It is clear that while the vast majority of measured samples are within one log difference of one another for the different methods, a small set of enumerated samples show a greater than 1.5 log difference in the titer values between the three algorithms. In these instances where the different algorithms lead to disagreement between the values computed by

the different approaches, the images were flagged for follow-up investigation by a qualified lab worker with training in manual enumeration and titer calculation to confirm the correct value and add corrections to the database.

In addition to the between-methods correlation for continuous evaluation of the pipeline's consistency, a collection of 768 automated counts was compared to human manual counts from two analysts (see Supplementary Fig. 4). Of the 768 samples, 53 were identified that had at least a 1.5 log discrepancy between the two analysts or between an analyst and the automated counting method. Each of the 53 samples was manually reviewed by a third analyst who provided an additional data point to determine a reportable count, which was then converted to a bacterial titer, see (Fig. 5d). In all cases the third analyst calculated a titer within a $0.5 \log_{10}$ range of either both manual analysts or one manual analyst and the automated method, providing additional confidence in the reportable counts. With respect to the refereed counts, the automated counting method was determined to be accurate (within 1 log) in 49 of the 53 reviewed samples (92.5%) (Supplementary Fig. 4). Analyst 1 was accurate in 10 of the 53 reviewed samples (18.9%), and analyst 2 was accurate in 31 of 53 samples (60.4%). Across all 768 samples, the automated counting method was accurate in 99.5% of samples, whereas analysts 1 and 2 were accurate in 94.4% and 97.1%, respectively. Overall, the watershed counting method provided counts with strong correlation and a greater agreement to corrected manual counts than either analyst (Fig. 5d, Supplementary Fig. 4). In sum, this pipeline provides automated counting and quality control flagging, allowing scalable assessment of bacteria and phage titers.

### Automated phenotypic assessment of high-titer manufactured LBP-EC01

After evaluation of the top 28 cocktails through CFU reduction experiments, six phages were selected to compose LBP-EC01. Each phage was subjected to process and formulation development to develop the manufacturing conditions required for generating high-titer phage material while removing impurities such as endotoxin. After suitable conditions were established, each phage was manufactured under Good Manufacturing Practices (GMP) conditions. Manufactured phages were combined into a final cocktail with an aggregate titer of $2 \times 10^{11}$ PFU/mL, which maintained titer within 0.5 $\log_{10}$ for 24 months at 4 °C.

Manufactured LBP-EC01 was tested in the optical-density-based assay against each of the 356 strains in the UTI Clinical Panel. Consistent with previous experiments of non-GMP material, LBP-EC01 exhibited activity against 96.4% of strains (Fig. 6). AUC ratios were heavily weighted towards the maximum detectable suppression. As a final assessment of LBP-EC01, this material was tested in a CFU reduction assay across the 356 strains. Each strain was exposed to a buffer control or three different titers of LBP-EC01 and incubated at 37 °C in a shaking incubator. Using the automated enumeration pipeline, the bacterial titer of each reaction was assessed at 0, 4, and 24 h. At 4 h, in the control reactions that lacked phage, bacterial samples grew to an average concentration of $1.2 \times 10^9$ CFU/mL. In contrast, the average concentration of bacterial strains treated with the highest concentration of drug product was reduced to $3.6 \times 10^3$ CFU/mL (Fig. 6a,b). 94.0% of strains were reduced by at least 4 logs compared to the control, and 97.2% of strains by at least one log. E. coli CFU reduction was dose dependent, with 88.8% and 77.0% of strains reduced by at least 4 logs when treated with $5 \times 10^9$ PFU/mL and $1 \times 10^8$ PFU/mL of drug product, respectively. At 24 h post-phage treatment, control reactions lacking phage but containing bacteria grew to an average concentration of $2.6 \times 10^9$ CFU/mL compared to $5.5 \times 10^4$ CFU/mL in the highest titer LBP-EC01-treated samples (Fig. 6c, d). 75.5%, 60.7%, and 40.7% of strains were reduced by at least 4 log in reactions treated with $2 \times 10^{11}$, $5 \times 10^9$, and $1 \times 10^8$ PFU/mL, respectively.

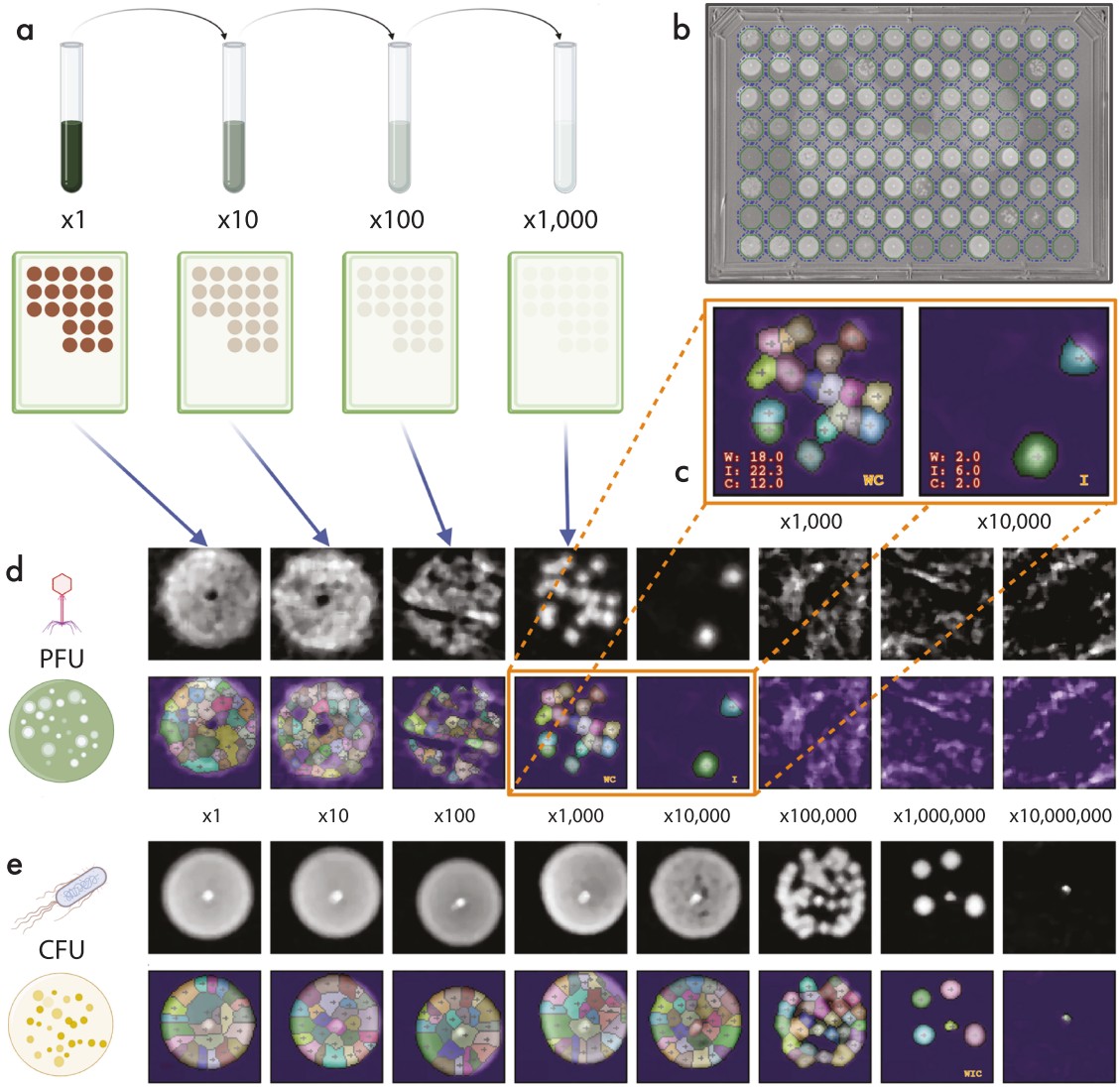

**Fig. 4 | Automated PFU and CFU enumeration of microbiological reactions.**
**a** Liquid cultures of samples containing bacteria or phage are serially diluted from the neat stock 7 times at factors of 10x and spot plated onto Nunc OmniTray Single-well plates in aliquots of 2 $\mu$L per spot. **b** An example image of the light-to-dark contrast for a spotted plate after subtracting a reference image, but prior to full background subtraction. The contrast image is inverted for CFU samples to allow the same enumeration algorithms to be applied for both sample types, even though plaques have lower optical density than the surrounding bacterial lawn, where they form, while colonies have higher optical density than the surrounding clear agar.
**c** An example image showing the resulting counts for the Watershed (W), Circle (C),
and Integral (I) enumeration methods. **d** A fully enumerated spot titer series for a PFU sample, which presents greater challenges due to the decreased contrast relative to background for faint plaques, with calculated titers for each method of $W_{titer} = 9.0 \times 10^6$ PFU/mL, $I_{titer} = 3.0 \times 10^7$ PFU/mL, $C_{titer} = 6.0 \times 10^6$ PFU/mL. **e** A fully enumerated spot titer series CFU sample with calculated titers for methods of $W_{titer} = 2.5 \times 10^9$ CFU/mL, $I_{titer} = 9.0 \times 10^9$ CFU/mL, $C_{titer} = 2.0 \times 10^9$ CFU/mL. Numbers outside of images indicate the dilution factors from the neat sample. Color differences indicate uniquely identified colonies or plaques. Created in BioRender. Tynes Hammack, A. (2025) https://BioRender.com/n82ti06.

## Automated phenotypic assessment of isolates from a Phase 2b clinical trial

A Phase 2b clinical trial was designed to assess the efficacy of LBP-EC01 concomitant with trimethoprim/sulfamethoxazole (TMP/SMX) in female patients with acute, uncomplicated UTIs (NCT05488340). The trial was composed of two parts: 1) an unblinded and open-label lead-in study to assess dosing of LBP-EC01 in 39 patients, and 2) a controlled, double-blind study to assess LBP-EC01 efficacy in up to 288 patients. Part 1 of the trial completed[1] and was used to select the dosing regimen for Part 2. As part of the exploratory analyses in the trial, bacterial strains were isolated from urine and sequenced. In total, 166 *E. coli* isolates were collected from urine samples throughout the study time points, irrespective of being in a formal analysis population. Each isolate was tested for LBP-EC01 sensitivity via the double agar overlay assay[20]. Genomic analysis of sequencing data from each isolate

confirmed the presence of 38 unique *E. coli* strains across all patient populations and time-points. Of the 38 strains, 37 were sensitive to the phages in LBP-EC01 (97.4%). Each strain was also tested via the optical-density-based assay, and the distribution of the resulting AUC ratio scores was compared to the distribution of scores obtained from screening against the full UTI Clinical Panel (Fig. 7a). The distributions were highly similar, with no statistical difference between the two strain collections, with 35/38 (92%) falling below the AUC threshold. Similarly, each of the *E. coli* isolates identified from urine samples prior to treatment was assessed via the CFU reduction assay and the auto-mated enumeration pipeline. Again, there was no difference in the distribution of CFU reduction efficacy in strains isolated from the Phase 2b clinical trial in comparison to our UTI Clinical Panel (Fig. 7b). Finally, we assessed the sensitivity of all 166 *E. coli* isolates to LBP-EC01 by plaquing and optical density time course phage efficacy assays. For

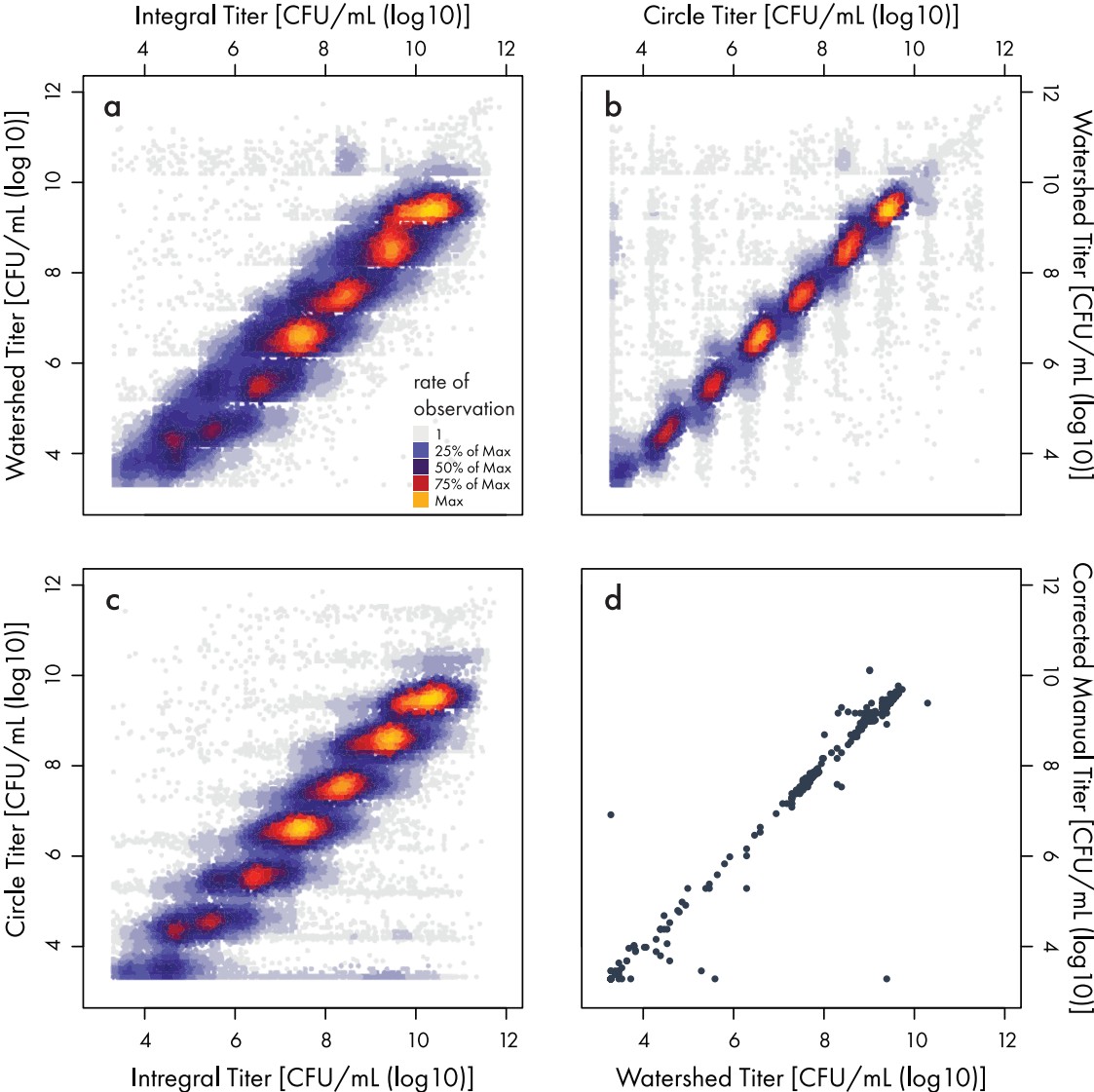

**Fig. 5 | Comparison of counting methods and throughput summary. a–c**
Heatscatter comparison between Watershed, Circle, and Integral methods, showing the correlation in the respective Colony Forming Unit (CFU)/mL on a $\log_{10}$ scale. The color scale increases from light gray to light orange and indicates the number of samples with matching $\log_{10}$ titer within a given region of comparison. The gaps between the different orders of magnitude represent the fact that the lower limit of counting is set to 3 (C/P)FU to account for artifacts of the spot plating method, while for colony counts $\geq 20$, colonies begin to overlap and become indistinguishable. These gaps can be readily eliminated, at increased consumables costs, by including intermediate dilution factors or through the use of larger spot (i.e., dispensed) volumes. **d** A small-scale comparison of counts by humans versus the quantized Watershed counting method. For discrepancies between manual counts between different analysts, a third tie-breaker assessment was manually performed to obtain the corrected manual counts.

each patient, we compared the genome assemblies of urine-derived *E. coli* isolates collected pre-treatment and across a 34-day period after initial treatment. Because some patients harbored multiple *E. coli* strains, we grouped isolates by sequencing type within each patient. Resistance to LBP-EC01 was defined as a loss of phage efficacy within the same sequencing type over time—either a >1-log reduction in plaquing efficiency or a shift from susceptible to resistant classification in the optical density assay (AUC ratio threshold = 0.65). Using this approach, we did not observe the development of genetic resistance to LBP-EC01 in any patient who enrolled in Part 1 of the study.

## Discussion

Phage therapy as a treatment for multi-drug resistant infections and microbiome dysbiosis has experienced a resurgence in recent years, and a number of clinical trials and Emergency Investigational New Drug studies have been conducted[17,21–24]. Several recent trials have failed to demonstrate efficacy[16], potentially due to insufficient host range or resistance prevention across a diverse set of strains. Combining ever-improving technologies such as high-throughput automation, artificial intelligence, and synthetic biology may enable the development of fixed phage cocktails that can treat patients in a scalable and cost-effective manner. To this end, we have developed an automated platform composed of the following operations: 1) Definition of a target bacteria panel, 2) Discovery and purification of phage from environmental samples, 3) Genotypic and phenotypic assessment of individual phages, 4) Synthetic engineering of phages, 5) Prediction of phage cocktail activity, and 6) Efficacy assessment of phage cocktails in vitro and in vivo. Herein, we focused largely on the methodology used to assess in vitro phage and phage cocktail activity, leaving the remaining topics to follow on publications.

This study describes a preclinical approach for characterizing phage-bacterial host interactions to inform phage cocktail development. The integration of automated liquid handling, image-based phenotyping, and cloud-based analysis tools enabled systematic

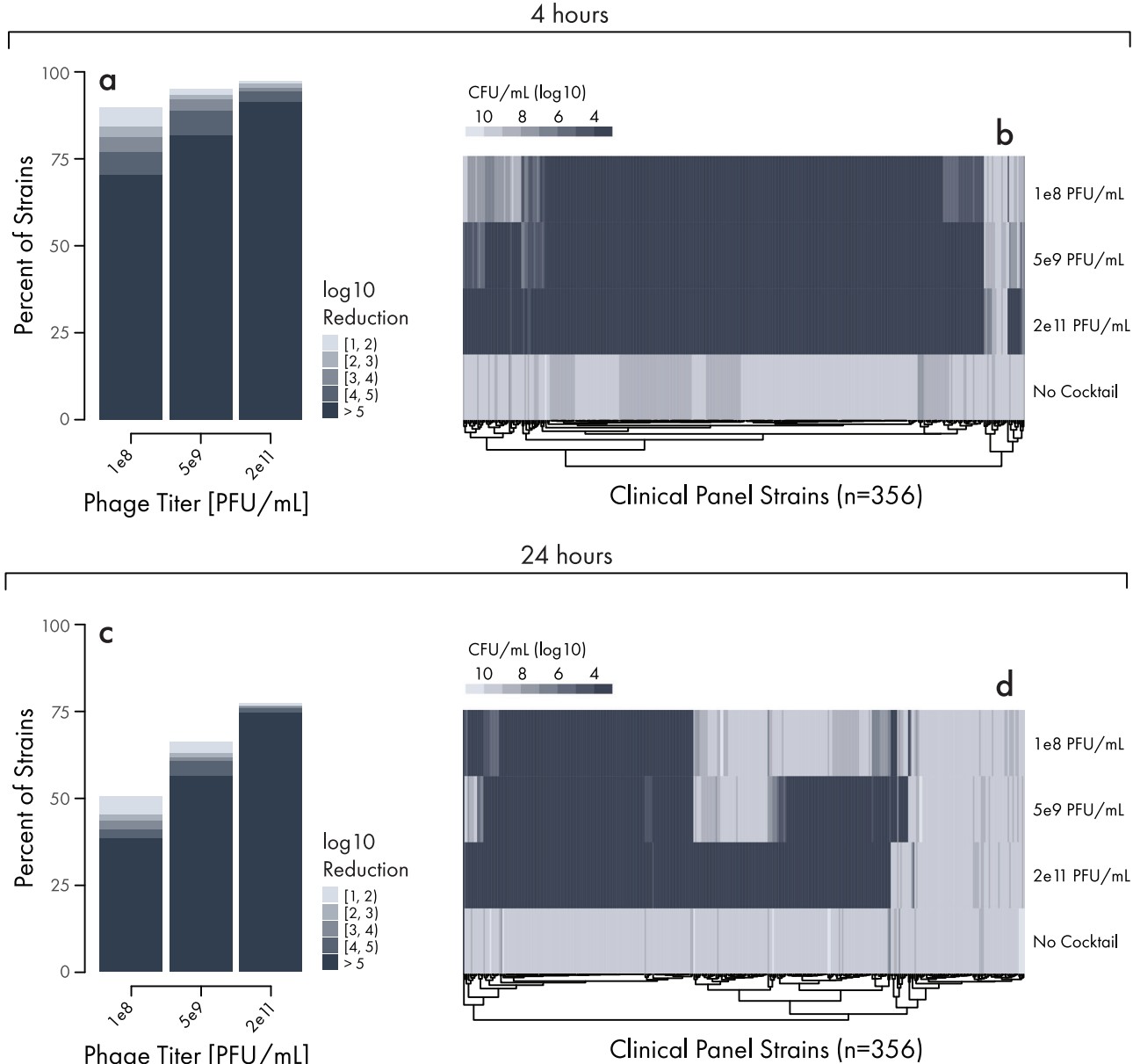

**Fig. 6 | Automated CFU Reduction of LBP-EC01 against Clinical Panel of UTI isolates.** Stacked bar chart showing the percentage of strains with varying Colony Forming Unit (CFU) reduction values 4 h (**a**) or 24 h (**c**) after phage cocktail exposure (interval notation is used for legend). Isolates were exposed to each of three aggregate exposure titers of the phage cocktail or a vehicle control represented in Plaque Forming Units (PFU)/mL. CFU reduction is calculated as the difference between the $log_{10}$-transformed bacteria titer values in reactions with and without phage cocktail exposure after 4 or 24 h. Corresponding heatmaps showing the CFU/mL values of each isolate on the Clinical Panel 4 h (**b**) or 24 h (**d**) after cocktail exposure. Darker colors correspond to lower CFU/mL values and improved cocktail efficacy. Darker colors in vehicle control indicate strains with slower growth.

screening of phage-bacteria combinations. These methods were used to identify a candidate phage cocktail that exhibited broad in vitro activity across a panel of UTI isolates. A large challenge in the phage therapy field is understanding the translatability of pre-clinical assays to clinical results, which will require statistically powered trials and extensive assessment of isolates collected from those trials as phage therapies progress through clinical trials and enter into broader therapeutic use. In addition to the assay itself, several other parameters can affect phage efficacy, including phage and bacteria titer, bacteria growth state, and environmental conditions of the reaction[25,26]. In the absence of a clear predictive assay and given the likelihood of disparate bladder environments across patients[27], we took an ensemble approach to the assessment of individual phages and cocktail performance. Automation of these assays was key to

enabling a thorough assessment of a multi-variate design space. Even so, additional assays such as organoid models may be needed to accurately represent clinical environments.

In a similar ensemble approach, the computer vision pipeline leveraged three distinct methods for phage plaque and bacterial colony counting and was assisted via automated quality control flagging for human intervention. Human errors are well known to contribute to inaccuracies in scientific measurement and require specific diligence within the context of healthcare and pharmaceutical development, given their potential high impact on patient outcomes[28]. The development of automated tools with reproducible and systematic forms of error is a key tool to reduce the impact of errors in planning and execution due to operator fatigue, knowledge gaps, or distraction, which are highly relevant to repetitive high-focus tasks such as colony

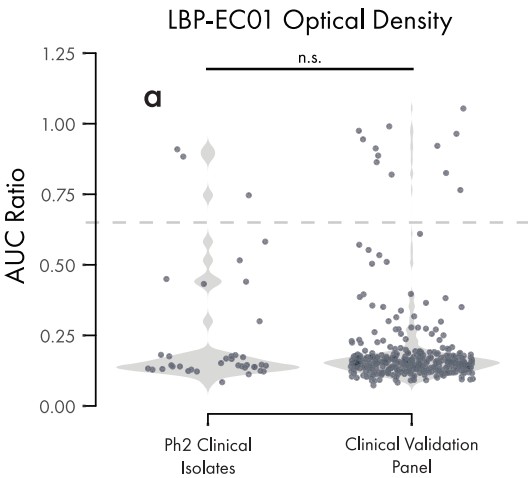

**LBP-EC01 Optical Density**

**LBP-EC01 CFU Reduction**

**Fig. 7 | Cocktail in vitro efficacy against Phase 2 Clinical Isolates.** Violin plots showing the optical density time course (**a**) or CFU reduction (**b**) based host range of LBP-EC01 on the UTI Clinical Panel compared to isolates collected from the lead-in portion of a Phase 2 (Ph2) clinical trial. Optical density values are plotted as AUC Ratios, and the dotted line indicates the threshold AUC ratio of 0.65. CFU reduction values are plotted as the $\log_{10}$ fold reduction of CFU/mL in samples treated with LBP-EC01 compared to a control buffer. Differences in AUC ratio and CFU reduction values between the two strain panels were not statistically significant as determined by the Welch two-sample $t$-test ($p$-value 0.13 and 0.47, respectively).

and plaque enumeration[29]. We observed a significantly higher reliability of enumeration with the computer vision pipeline than the results of trained human analysts. Although highly accurate alone, a key benefit of the methods in this work was the 'Centaur' counting approach[30], where computer vision results are assisted by human intervention when automated counting methods are in disagreement. This approach provided a well-curated and audited novel dataset that enabled the correction of repeated errors present in the automated counting. A survey of the flagged outliers showed that the image processing pipeline can struggle to accurately enumerate plates with very faint plaques, as well as spotting regions that are near the edges of the plates next to the 45° corner notches. This audited dataset can be used to train artificial intelligence models to increase speed or accuracy, and we have had early successes with neural network models trained across this extensive imaging dataset.

The phage cocktail developed as part of this work, LBP-EC01, has been evaluated in the dose-finding portion of an ongoing Phase 2 clinical trial[1]. In an exploratory endpoint of the non-controlled portion of the trial, urine bacteria levels were assessed in the pharmacodynamic evaluable population. This population included 16 patients containing E. coli at baseline, of which 11 exhibited TMP/SMX resistant E. coli at baseline. 14 of 16 patients exhibited microbiological cure at the day 10 test of cure, and 16 of 16 patients demonstrated absence of clinical symptoms throughout the study, which ended 34 days after the initial treatment. The double-blinded and controlled portion of the trial will be used to assess the safety and efficacy of LBP-EC01 in a larger patient population. Notably, in both the optical density time course phage efficacy and CFU reduction assays, we observed that the distribution of antibacterial efficacy measurements was similar between strains isolated on the UTI Clinical Panel and strains recovered from patients in this portion of the trial. The consistency between in vitro results from the training panel and patient isolates suggests that the panel design may capture relevant strain diversity; however, further in vitro characterization of isolates obtained from the remainder of the Phase 2 trial will be of interest, given the relatively limited sample size of Phase 2 isolates reported here. Of the initial 14 patients with microbiological cure, all 14 were sensitive to LBP-EC01 in all three tested assays. In the remaining patients, one patient contained isolates sensitive to LBP-EC01 and exhibited a 100-fold reduction in bacterial burden, but did not clear the infection by day 10. The final patient contained an isolate insensitive to LBP-EC01 and resistant to TMP/

SMX. This patient maintained a consistent bacterial burden throughout the trial. Initial statistical correlations were hindered by a paucity of isolates insensitive to LBP-EC01; however, further analyses will be of significant interest at the conclusion of the Phase 2 trial, consisting of a larger patient population.

In conclusion, the pre-clinical characterization efforts highlighted in this study underscore the importance of thorough evaluation processes in the selection of phage cocktails. Preliminary data from the clinical trial indicate a correlation between in vitro assay results and observed microbiological or clinical responses, though further study in larger patient populations is needed to confirm this relationship. The integration of high-throughput assays, predictive models, and synthetic biology will be crucial in refining this therapeutic approach and facilitate unlocking the promise of phage therapy in the treatment of multidrug-resistant infections and dysbiotic microbiomes.

## Methods
### Automated strain panel generation
To ensure consistency in bacterial input titers and growth state across different assays, each bacterial strain on the UTI Clinical Panel was normalized and aliquoted into single-use tubes. Bacterial strains generally arrived in frozen cryovials, and upon intake, a sterile 10 μL loop was used to inoculate 1 mL of LB media in a deep well block. 48 strains were added in a checkerboard pattern per deep well block and were incubated in a shaking incubator at 37 °C for 4–16 h. Deep well blocks were then diluted and spotted using the liquid handling method described in the Automated Colony and Plaque Enumeration Assay section. Individual colonies were picked using a sterile 10 μL loop and added to a 14 mL sterile culture tube containing 5 mL of LB media or a 50 mL sterile, vented cap tube containing 20 mL of media. Tubes were then incubated with shaking at 37 °C overnight (12–16 h). All culture tubes were then loaded onto a Hamilton VANTAGE 2.0 liquid handler for normalization and aliquoting. A 200 μL aliquot of each culture was added to a 96-well plate, and the optical density of each sample was read at 600 nm and checked to ensure cultures were in the linear range of detection. Subsequently, the liquid handler normalized each culture to an OD of 0.02 in 4 identical deep well blocks, added glycerol as a cryoprotectant to a final percentage of 20%, and mixed the resulting samples. 180 μL was then transferred via the 96-channel pipettor to racks that each contain 96 pre-barcoded tubes with a 300 μL volume capacity from Hamilton Storage Technologies. Racks were loaded into

a SAM HD −80 °C automated biobank, also from Hamilton Storage Technologies. This biobank has an 80,000-tube capacity and contains a tube picking module that allows custom organization and recall of individually banked tubes. This module was used to obtain smaller sub-panels of strains for different assays while maintaining consistency in starter material. Multi-Drug Resistant (MDR) strains were defined as resistant to at least one antibiotic in at least 3 distinct classes. All isolates were collected from female urinary tracts. In most cases, these isolates were from urine. A minority of isolates were collected from catheters ($n = 17$) or were labeled by the providing vendor as Urinary Bladder ($n = 3$).

### Automated optical density time course phage efficacy assay

The optical density time course phage efficacy assay was run on a custom-designed Hamilton VANTAGE 2.0 with A, B, C, and D modules. Module A refers to the pipetting deck, whereas Module B is the logistics cabinet that supports the pipetting deck. Module D is the rear cabinet containing the incubators and plate readers. Positioned between Modules A/B and D, Module C is the track gripper or robotic arm that facilitates plate movement across the system. The assay was coded in INSTINCT V software. Plate additions, incubations, and plate readings can occur simultaneously through this scheduling software. User inputs for a unique experiment identifier, number of phages, number of optical density reads, phage identity and titer input file, bacteria identity input file, incubator temperature and shaking speed are recorded in a run metadata file. Serial numbers of all instruments used in a given run and the time of run initiation were also recorded in this file.

384-well plates containing media, phage, and bacteria were generated by the pipetting module of the liquid handler (Supplementary Fig. 2). Sterile, optically clear, non-surface-treated, and lidded microtiter plates were used and were stored in the Entry/Exit module of the instrument upon run start. Plates are processed in batches of two and are moved to an on-deck carrier position for liquid addition. The 96-channel pipettor then adds 40 μL of media to each well of the 384-well plate. Subsequently, individual spanning channels are used to add 5 μL of phage via a continuous surface dispense. Finally, the 96-channel pipettor adds 5 μL of bacteria from 4 racks of 96 pre-barcoded tubes and mixes the 50 μL reaction volume. 384-well plates are re-lidded and transferred to a StoreX Liconic shaking incubator with temperature control. Shaking rpm was varied for different bacteria species and was set at 250 rpm for E. coli. This shaking speed was selected based on initial growth optimization experiments for a random selection of E. coli strains normalized to a starting optical density of 0.02, in which various rpm speeds between 100 and 800 rpm were studied. Of these, 250 rpm was selected because it optimized for maximal area under the bacterial growth curve, while minimizing the speed. Minimizing speed was an optimization objective, as it reduces both the risk of cross-well contamination within the 384-well plates and the long-term stress on the automated shaking incubators. Orbital shaking amplitude was 2 mm. After the plate addition to the incubator, the time frame of plate completion was recorded in the run metadata. After completion of all 384-well plates, scheduled transfers move plates from shaking incubators to a Biotek Powerwave HT plate reader. Before each read a 2 s plate shake was instituted to avoid cell settling during plate transfer. Wavelength dependent OD readings were taken at 600 nm and were written to an XML file in the run output folder with a corresponding time stamp. After the collection of all reads across all plates, typically between 20 and 40 h, the time of run completion was recorded.

Post-run schedulers move metadata and raw data captured by the instrument to cloud storage, which are then passed to CI/CD-based analysis software pipelines to ensure reproducible code execution through software version control and containerized computing environments (see Automated Colony and Plaque Enumeration Assay). Code for XML and run metadata parsing was generated in Python, and

quality control reports were generated in R-based Jupyter notebooks triggered by `Papermill`. AUC measurements were taken by integrating under the optical density growth curve between the 3 and 20 h time points. The AUC ratio value is calculated by dividing the AUC measurement of the phage-treated growth curve by the AUC measurement of the control growth curve.

### Automated colony and plaque enumeration assay

500–1000 μL mixtures of media, phage, and bacteria were generated in deep well blocks manually or with liquid handlers and were stored in shaking incubators at 37 °C. Phage and bacteria sample identifiers and concentrations were recorded in input files uploaded during auto-mated run initiation. At pre-determined timepoints, dilution and spotting of the reaction plate were triggered using a Hamilton VAN-TAGE instrument pre-loaded with 50 μL nested, sterile tip racks, sterile microtiter dilution plates, and barcoded agar plates loaded in the Entry/Exit elevator module. An aliquot was taken from the reaction plate via the work cell's 96-channel pipettor, and the reaction plate was returned to the shaking incubator. For colony enumeration, aliquots were taken directly into serial dilution, and for plaque enumeration, aliquots were passed through a 0.22 μm polyethersulfone (PES) filter. Each deep well block was serially diluted 1:10 into 1x phosphate-buffered saline (PBS) across 7 additional microtiter plates using the 96-channel pipettor. The neat plate and each of the 7 dilution plates were subsequently used to dispense 2 μL spots of each sample plate onto delidded, rectangular agar plates. For colony enumeration, neat and diluted samples were spotted on LB rectangular agar plates. In contrast, samples used for plaque enumeration were spotted onto a double-agar overlay. To generate this overlay, 100 μl of overnight bacterial culture was added to 11 mL of LB low-salt top agar (Teknova L7570) and thoroughly mixed by inversion. The mixture was then poured evenly across an LB rectangular agar plate and allowed to dry for 30–60 min. Barcode scanners integrated into the liquid handler were used to record the source plate and time stamp for each dispense. Scheduling software was used to control the dispense of each plate, a variable dry time under HEPA filtered air, restoration of the plate lid, and transfer back to the Entry/Exit module or the Liconic incubator on a static setting. Run input files, time stamps, and other relevant metadata were stored in a folder with a unique experiment identifier upon run completion. Agar plates were allowed to dry and were incubated upside down in a static incubator for an additional 12–16 h to allow colony formation. Finally, plates were imaged on a robotic work cell equipped with plate storage, a light table and IDS UI-3060CP Rev. 2 camera featuring a Sony IMX174LLJ monochrome imaging sensor with 5.86 μm pixels possessing a large full well depth of 31,278 electrons, and 12-bit ADC (4096 levels) readout that provides a large number of gray levels required to discriminate faint phage plaques from their surroundings. Metadata from the work cells that performed the assay, as well as the raw images, were ingested into cloud storage for further analysis.

The enumeration of the serially diluted sample was performed using a custom image analysis pipeline, written in Python, using a high-performance parallel cloud architecture orchestrated using Apache Airflow for large-scale task scheduling and Dask for parallelism within each task-specific allocated compute node. Each step in the enumeration pipeline was launched through the submission of configuration parameter files to a GitHub repository that, in turn, triggered Google Cloud Build processes to perform the actual computing tasks on pre-compiled container images. This overlay of GitHub with computation on cloud compute architecture enables highly reproducible computing with clean provenance and audit trails for all of the operations performed on the data. The combination of the git repository to version the evolution of the code base, the container identifiers to record the specific computing environment, and the cloud logging databases to provide long-term storage

of all of the run-specific outputs or errors leads to a detailed record of all samples processed through the pipeline. Additionally, the use of cloud storage with well-defined retention policies enables the archival of metadata and computed measurements by leveraging the best current enterprise practices in multi-site and multi-method data backup through cloud automation.

Prior to evaluation by the three separate image enumeration algorithms, the images were background subtracted and aligned to a uniform coordinate system to account for slight variations in the position of the plates relative to the camera, or for adjustments in the camera zoom factor under normal operation due to machine maintenance or analyst intervention. The background subtraction and image optimization methods will be detailed in a separate following manuscript, given their general applicability to a wide variety of image processing tasks outside the scope of phage discovery and characterization. The major elements for the background subtraction consist of the following steps: 1) registration of the image to physical coordinates in millimeters using alignment marks from the plate imaging station on the robot, 2) subtraction of an average blank plate image from the image to be enumerated, 3) cropping of the image to the region within the plate boundaries, 4) subtraction on a pixel by pixel basis of the minimum intensity values at each $x, y$ coordinate from all plate images within the enumeration plate image series, 5) median filtering of the image using a $5 \times 5$ median blurring kernel, 6) defining a triangular mesh that samples pixels in locations known to be outside the spotting locations in order to create a low spatial frequency model image of the background variation in the agar optical density or other light scattering artifacts such as condensation, 7) subtraction of this background model image from the image created by the preceding steps to create the final background subtracted image that will be passed to the enumeration algorithms. A representative background subtraction and image normalization processing series is shown in Supplementary Fig. 3, demonstrating the importance of removing non-uniform background features prior to analysis for counting. Following background subtraction and coordinate alignment, each of the spotted areas was segmented into smaller images for parallel processing across the cloud computing cluster for the Watershed and Integrated counting methods. Whole plates were processed for the Circle counting method.

The Watershed method leverages a marker-based modification to the original watershed algorithm[31,32], which involves treating a gray-scale image as a topographic map, and seeks to segment based on the location of basins within the topography. The intensity of light transmission through agar growth plates is proportional to the optical density at a given position on the plate, and the OD increases with the number of bacteria in the surface layer of the agar. Given these aspects, the OD imaging measurements for phage plaques formed in the agar map excellently onto topological basins. For bacterial colonies, the image is simply inverted prior to the watershed calculations in order to convert the increased OD in regions containing colonies into basins instead of hilltops. The markers for each watershed were chosen by finding the local minima using a minima finding image filter with a radius commensurate with the average colony or plaque size of approximately 1 mm. The number of distinct watersheds found within the 9 mm region of a given spotting area is the count provided by this method.

The Integrated method uses a simple normalized sum of the total image intensity within the background-subtracted and sub-segmented image encompassing only the immediate 9 mm square area surrounding a single spot out of the full $8 \times 12$ plate of 96 spots, using the formula

$$I = A \sum_x \sum_y \text{intensity}(x, y), \tag{1}$$

where $x$ and $y$ denote the horizontal and vertical coordinates of the image, *intensity* is pixel intensity at the image position in arbitrary units, and $A$ is a normalization coefficient chosen by statistical analysis of calibration runs such that the total range of $I$ roughly matches the integer counts provided by manual counting, Watershed counting, and Circle counting of the same regions.

The Circle method uses the circle Hough transform[33] to find circles within the image at characteristic radii typical to individual phage plaques or bacterial colonies, approximately 0.5 to 2 mm, as well as for full zones of clearing (ZOCs) where the entire spotting area of bacterial lawn is eliminated by the phage, or the bacterial colonies are fully overlapping and indistinguishable, which is approximately 8 mm for the 2 $\mu$L volumes dispensed routinely in the assays. If a ZOC is found at a given dilution, the maximum distinguishable count of 30 is set for the region; otherwise, the number of smaller circles detected within the region is reported as the count for that region. 30 was chosen as a value, as it is typically too challenging to count 30 colonies or plaques within the diameter of a 2 $\mu$L spot.

The counts from the different methods are combined into separate vectors to represent the counts across the full dilution series for a given sample for each of the enumeration methods. These vectors are then converted into a titer value for each separate method that represents the series according to the following algorithm. The titer calculation algorithm iterates through the list of counts and corresponding dilution factors, ordered by increasing dilution factor, and selects the dilution factor of the first item in the list with a count value that is less than one-half of the preceding element in the list. In the event that no such greater than two-fold fall-off is found, the first item in the list is chosen as the count and dilution factor to represent the series. The titer is then calculated by the following formula

$$T = \frac{(W|I|C|M) \times F}{V}, \tag{2}$$

where $T$ is the resulting titer in (C/P)FU per mL, $V$ is the dispensed volume, $W$, $I$, $C$, or $M$ are the Watershed, Integrated, Circle, or Manual counts depending on the method, and $F$ is the dilution factor for the chosen element in the series. The $\log_{10}$ titer is calculated as the base ten logarithm of the titer as $T_{\log 10} = \log_{10}(T)$. Generally, for the measurement of relative variability in an assay that spans many orders of magnitude, statistical comparisons using a logarithmic basis have favorable mathematical properties[34].

## Clinical trial strain assessments

Strains were isolated from patient-provided samples as part of an ongoing two-part Phase 2 controlled trial named ELIMINATE (NCT05488340). Informed consent was obtained from patients providing urine cultures before trial enrollment. The study protocol and informed consent form were approved by the central Advarra Institutional Review Board (see Ethics Statement). Urine samples were collected by clean catch or catheter sampling across a 34-day period[1]. *E. coli* isolates were collected by International Health Management Associates (IHMA) and confirmed via matrix-assisted laser-desorption ionization time-of-flight (MALDI-TOF). Isolates were stored at −80 °C in cryovial tubes. Upon tube receipt, cultures were struck via sterile 1 $\mu$L loops, and allowed to incubate overnight at 37 °C. Subsequently, a single colony was picked and used to inoculate a 14 mL sterile culture tube. Cultures were normalized and aliquoted as described in the Automated Strain Panel Generation section.

A double agar overlay assay was performed by mixing 150 $\mu$L of an overnight culture of a given bacteria strain with 11 mL of soft agar, which was then poured over a rectangular agar plate and allowed to solidify for approximately 15 min. 1:10 serial dilutions of individual phage or the LBP-EC01 cocktail were then dispensed on top of the solidified agar in 2 $\mu$L spots. Spots were allowed to dry for 20–30 min

before incubating at 37 °C overnight to allow formation of a bacterial lawn. Sensitivity to individual phages or to the cocktail was defined as observation of individual plaques or zones of clearing in the phage-exposed regions of the plate.

To sequence clinical isolates, 1 mL of overnight culture of each strain was pelleted at 4000 g for 10 min, and the supernatant was removed. DNA was extracted using the MagMax Microbiome Ultra Nucleic Acid Isolation Kit (Thermo-Fisher). Isolates from urine samples were collected pre-LBP-EC01 treatment in the trial and were sequenced on a PacBio Sequel IIe using the SMRTbell prep kit 3.0. All other isolates were sequenced on an Illumina MiSeq using the Illumina DNA Prep library kit. Sequencing data from E. coli isolates were examined for each patient using whole genome alignment and review of sequencing type designations. E. coli strains were designated as unique in this study if they were from different patients or had a distinct sequencing type from another isolate within a given patient.

### Statistics and reproducibility
Phage and phage cocktail efficacy were evaluated using a panel of 356 E. coli strains. This panel was established from genomic analyses of over 10,000 E. coli sequences, with representatives selected to capture broad biological and genetic diversity among clinical isolates. Each phage-bacteria combination was tested in 1–3 biological replicates to assess reproducibility, and the large number of strains ensured a comprehensive evaluation of phage efficacy across diverse genotypes.

For enumeration analyses, a random subset of 500,000 colony counts was selected to enable clear visualization of titer distributions and method performance comparisons. Operators were blinded to the counts of the other operator and the computer vision model. Colony-forming unit (CFU) reduction experiments were performed in singlicate across the full 356-strain panel, with assay breadth compensating for replicate number to provide robust, population-level insight into phage activity. Phage cocktail efficacy on this panel was subsequently compared to all genomically unique clinical isolates from the Phase 2 trial using a Welch two-sample $t$-test to assess consistency of host range coverage.

For the simulation of cocktail combinations for simple additive host ranges, the number of simulated combinations was capped at $\binom{170}{2} = 14,365$. The swarm and violin plots were capped at 340 samples of the total collection of simulated cocktails for the sake of visibility during plotting. These values were sufficient to illustrate the trends in the performance of naive algorithms for cocktail prediction relative to the final results from the full proprietary methods employed for clinical phage cocktail selection, as well as comparison to the performance against isolated clinical strains.

### Ethics statement
This work utilized bacterial isolates collected from patients with recurrent urinary tract infections. The study protocol and informed consent form were approved by the central Advarra Institutional Review Board, which is compliant with all applicable laws and regulations and ethical principles for human subject research, including The Belmont Report, Nuremberg Code, and Declaration of Helsinki. All eligible patients were informed about the study procedures and risks and provided written informed consent before enrollment. The informed consent forms included the following language regarding the collection of clinical isolates: Individual bacterial or phage isolates recovered from you may be retained for perpetuity and may lead to new tests, drugs, devices, or other products or services with commercial value.

### Reporting summary
Further information on research design is available in the Nature Portfolio Reporting Summary linked to this article.

## Data availability
Source data are provided with this paper and have been deposited in the Zenodo database under DOI: 10.5281/zenodo.17594564[35]. The raw data used in the development of LBP-EC01 are proprietary and are not available for general distribution. Source data are provided with this paper.

## Code availability
Provided automation code has been deposited in the Zenodo database under DOI: 10.5281/zenodo.17594564[35]. The remaining code used for analyses in this work is proprietary software and is not available for general distribution.

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

## Acknowledgments

This work was supported by Federal funds from the Department of Health and Human Services; Administration for Strategic Preparedness and Response; Biomedical Advanced Research and Development Authority (BARDA), under Contract No. 75A50120C00169 (PG). Work at the Molecular Foundry was supported by the Office of Science, Office of Basic Energy Sciences, of the U.S. Department of Energy under Contract No. DE-AC02-05CH11231 (ATH). We acknowledge Jones Microbiology Institute (JMI) and International Health Management Associates (IHMA) for providing bacteria strains. We also acknowledge Aaron Stepanek, Brandy Barber, and Megan Frisbee for assistance in the experimental setup.

## Author contributions

TJRP: Conceptualization, Formal Analysis, Software, Supervision, Writing A.T.H.: Conceptualization, Formal Analysis, Software, Supervision, Writing L.J.M.: Conceptualization, Supervision E.B.: Conceptualization, Investigation P.W.: Investigation N.H.: Methodology, Software M.K.Y.W.: Methodology, Software, Visualization N.C.: Investigation I.W.: Methodology, Software H.T.: Conceptualization, Supervision S.W.: Investigation A.T.: Project Administration C.J.P.: Methodology, Software E.D.: Investigation A.G.: Investigation D.G.O.: Conceptualization, Funding Acquisition, Supervision N.R.C.: Conceptualization, Funding Acquisition, Supervision P.G.: Conceptualization, Funding Acquisition, Supervision. All authors provided writing review and editing. All authors have consented to publication.

## Competing interests
