## [Transparent Peer Review file · Nature Communications]

High-throughput methods leveraging robotics and computer vision for the development of therapeutic bacteriophage cocktails

Corresponding Author: Dr Taylor Penke

Version 0:

Reviewer comments:

Reviewer #1

(Remarks to the Author)

The research article by Penke et al. on “High-throughput methods leveraging robotics and computer vision for the development and assessment of therapeutic phage cocktails” describes the automated phage screening techniques to improve the efficacy of phage selection in preclinical and clinical setting. Presenting this huge data set results are commendable. The idea of using modern technologies like robotics and computer vision in phage therapy is attractive, especially (methods) screening huge sets of pathogens and phages is impressive. Even though the efficacy of PT as a stand-alone therapy to treat AMR infection needs further investigation.

1. One hypothetical question is the feasibility of using this method in personalized phage preparations when infections are more complicated than *E. coli*.

2. How effective is this method in selecting phage cocktails?

[The combination of phages in cocktail should have different receptors to eliminate the development of resistant mutants.]

3. What is the rate of (detection) error? Is it 100% accurate? As this article is all about the methods, it is important to discuss the inaccuracies in the detection.

Minor:

1. Line no. 108: Are the isolates from urine samples? Need clarity.

2. Line no. 109: Check: 98% of strains or isolates?

3. Line no. 150: What are those phages belonging to?

4. Line no. 169: To be more accurate, the growth media is related to bacteria which eventually relates to phage propagation.

5. Line no. 246,247: LBP-EC01 is one of the 28 cocktails or different.

6. Line no. 288: It should be '38 stains'.

7. Line no. 303-327: Correct the typo in degree sign.

(Remarks on code availability)

Reviewer #2

(Remarks to the Author)

This manuscript contains some impressive results that move phage cocktail therapy toward clinical trials. The problem is that large sections of the paper currently read like a press release. The bulk of the potential contribution to the field is in the data itself on phage-bacteria interactions, which is not released, and the 'proprietary cocktail prediction model', which is not described and can't be replicated.

We think with some edits this could be a very nice paper. It is a powerful story that stretches all the way from the nuts and bolts of engineering a scalable data collection solution for the problem, using that data to solve the main barrier to application, and then applying it to a problem of clinical relevance. The edits that would strengthen the paper is describing how the data is acquired *and* how that data is used to add value.

We appreciated seeing the robotic methods you arrived at for gathering reproducible data on phage infection at high throughput. This is the most technically detailed section of the paper, which we appreciate. With slightly more information (we make some suggestions below), these methods could be replicated and would be very useful for other phage biologists.

Suggestions on edits to robotic methods for gathering phage infection data

Would the authors be willing to release their automation code? They are presumably saved as a Venus file. The code contains a lot of details that would be valuable for using this technique again, i.e. liquid classes that are used for different reagents, whether liquid handling steps have been optimized to avoid traversing over other plates to avoid contamination, etc.

Figure 1 contains renders of their instrument. It would be more helpful to have a photograph, a deck layout image, and/or a visual depiction of the method itself (see an example in figure 2a of <https://www.embopress.org/doi/full/10.15252/msb.20209942>)

Could the authors provide more details and the automated preparation of the plaque forming assays? At the moment, it is not clear how the PFU assays are prepared - as described later in the manuscript one would expect a double agar overlay method here to establish a bacterial lawn on some of the plates. For the PFUs, are the phages separated from the bacteria before having samples plated?

Suggestions on edits to methods for using phage data to predict effective phage cocktails:

Currently the extent of detail is this sentence: "We developed a proprietary cocktail prediction model that leveraged our database of phage and bacteria reactions to recommend cocktails for additional testing." Figure 2 shows histograms of cocktail effectiveness, the top cocktails and their effectiveness against UTI strains. We understand that you can't release this algorithm, but it would be valuable to find other ways to discuss this part of the paper with more nuance.

Could you show the performance of the proprietary method relative to simple control algorithms for selecting phage cocktails? For example, you should show Figure 2c for the proprietary cocktail selection method vs random vs another simple method (say, regression based). This would help other scientists understand whether they could use a simpler, less effective technique and still have it work, or whether they should expect to need to develop something more sophisticated if they wish to attempt this sort of technique.

How much better is the proprietary method? How many fewer experiments did it allow them to run? How iterative/closed loop was it? How did they handle the explore/exploit tradeoff? These are the fundamental questions that is of a lot of interest to the field broadly. A lot of the potential value of your paper is interpreting it as a specific scientific example of the value of robotics + data analysis in solving high-combinatorial problems in life science.

The dataset itself, if released, would be valuable: 421 phages to 356 E. coli, which is more than the current state of the art: 96 phages to 403 E. coli from Pasteur recently (data <https://zenodo.org/records/13831957>, and paper <https://www.nature.com/articles/s41564-024-01832-5>). If it's not possible to release the data, perhaps you could use your computational tools on these public datasets versus your private dataset

Other comments by line:

Fig 3 d and e: What do the different colours mean in this figure?

Line 47: Reference 5 estimates annual deaths attributable to AMR to be 1.91 million (1.56–2.26) and 8.22 million (6.85–9.65) deaths associated with AMR in 2050. Because of the context of the sentence which uses the language "an estimated 1.27 million deaths were attributed to bacterial antimicrobial resistance (AMR) [2], and this number is estimated to be as high as 10 million by 2050", I think that either the wording should be changed to show that the 10 million (from ref 4) is not referring to attributable deaths or the 1.91 million number should be given.

Line 70: "their favorable safety profile" - should have a reference

Line 138, 147, 347: Shaking speed of the plates is important due to the effect on aeration (Shaer Tamar and Kishony 2022) (see additionally ref 30-38 of this paper) - shaking speed should be mentioned in the main text and orbital radius should be given in methods.

In line 347, why was 250rpm chosen for E. coli? What data informed this decision?

Line 169 and 172: What conditions were tested? What components of the media were varied? How did this affect the growth of the different bacteria and phages?

Some have observed phages showing limited efficacy in human serum - was this tested?

Line 217: Figure 4 doesn't have any PFUs in the axis titles but the text here says that PFUs are represented in the data - this is confusing

Line 300: If including this line, then state what method was used to come to this conclusion and ideally show the data

Line 330: I could not find any information online about what Modules A,B,C, and D of the Hamilton Vantage 2.0 system are

(Remarks on code availability)

Some methods are described, but the code isn't linked in the methods unless we're missing something?

Reviewer #3

(Remarks to the Author)

(Remarks on code availability)

Not available

Version 1:

Reviewer comments:

Reviewer #2

(Remarks to the Author)

These edits have greatly improved the paper and we appreciate the detailed answers to all of our comments.

Thank you for going to the effort of using OCR to get the code out of the INSTINCTV software for others to read and for sharing the GIFs of the simulations! We hope this makes your methods more widely accessible.

Cocktail selection analysis: The addition of Extended Data Figure 9, comparing your methods to random and top-k selection algorithms, is very helpful. Given its significance in illustrating how your proprietary active learning algorithm enhanced exploration of the phage sequence space, this figure might warrant consideration for inclusion in the main text.

Citation Correction: On a second read, we followed the citation chain for "The frequency and recurrence of UTIs are a significant health burden and are associated with an estimated \$1.6 billion annual cost [12]". The only reference in 12 that seems to be for the \$1.6 billion number is to a paper entitled "Risk factors for second urinary tract infection among college women" (from the line: "The estimated annual cost of community-acquired UTI is approximately \$1.6 billion.[15]"). This paper is about how the sexual behaviours of college women from 1992 to 1994 affected their probabilities of getting a repeat UTI - very interesting but there is no mention of the \$1.6 billion number! We request this citation be removed before publication. Rather than attempting to replace it with another economic estimate, we suggest the human health burden data already presented is sufficient to make your case.

We wish your clinical trial success and hope your future work leads to greatly improved patient outcomes.

(Remarks on code availability)

The authors share Hamilton automation code for the three assays used in the paper. To help others who try to replicate or use the code, they share videos of the simulations of the experiments when run on their hardware and software. Furthermore, for those with different hardware and software setups they share the raw text of the automation code for each assay (outside of the hardware manufacturer's proprietary file format).

Response to Reviewers

High-throughput methods leveraging robotics and computer vision for the development of therapeutic phage cocktails

We are grateful to the editors and reviewers for their thoughtful and constructive feedback, which has significantly improved our manuscript. In response, we have made substantial revisions to enhance the clarity, transparency, and reproducibility of our work. Key additions include 1) all of the raw data used to plot figures in tabular (.csv) form, 2) public release of automation code used to execute the combinatorial optical density assay to support adoption by other researchers, 3) improved methodological detail for the requested assays, and 4) a detailed response to all reviewers' comments and questions. Furthermore, to address questions about our cocktail selection model, we included a comparative analysis demonstrating its performance relative to simpler algorithms. We also clarified methodological details, corrected errors, revised the tone throughout the manuscript, and updated figures and legends for improved readability. Below, we provide point-by-point responses to each of the reviewers' comments, with our replies highlighted in blue. We hope we have sufficiently addressed all feedback and welcome any further questions or suggestions.

REVIEWER COMMENTS

Reviewer #1 (Remarks to the Author):

The research article by Penke et al. on “High-throughput methods leveraging robotics and computer vision for the development and assessment of therapeutic phage cocktails” describes the automated phage screening techniques to improve the efficacy of phage selection in preclinical and clinical setting. Presenting this huge data set results are commendable. The idea of using modern technologies like robotics and computer vision in phage therapy is attractive, especially (methods) screening huge sets of pathogens and phages is impressive. Even though the efficacy of PT as a stand-alone therapy to treat AMR infection needs further investigation.

We thank the reviewer for their encouraging comments and appreciate the recognition of our efforts to integrate robotics and computer vision into phage therapy workflows. We also agree with the reviewer's note that while the methodological infrastructure described here is a critical step forward, additional work—including double-blind, placebo-controlled clinical trials—is essential to fully assess the efficacy of phage therapy as a stand-alone treatment for antimicrobial-resistant infections. Please see the following in-line comments to respond to the reviewer's questions.

1. One hypothetical question is the feasibility of using this method in personalized phage preparations when infections are more complicated than *E. coli*.

We assume here that the reviewer is referring to polymicrobial infections that may be due to either multiple species or multiple strains of a single species that are causing an infection. In these cases, it may be necessary to target multiple pathogenic entities rather than a single

entity. Although our company is not focused on personalized phage therapy, and rather, focuses on development of a single cocktail for the treatment of a given type of infection, we believe that other groups' development of personalized approaches could play an important role in the therapeutic landscape. As such, our methods can also be applied to personalized phage therapy development. Both the optical density-based assay and CFU reduction-based assay can be run with either a panel of bacterial strains from different patients or a collection of bacterial strains from the same patient. In cases of personalized phage therapy where a single or smaller number of strains are the target for a phage cocktail, our platform would enable a higher number of phages or phage cocktails to be assessed. In other words, the total number of bacteria x phage reactions per instrument run is fixed, but the number of phage or bacteria can be flexibly changed to address the research question.

2. How effective is this method in selecting phage cocktails?

[The combination of phages in cocktail should have different receptors to eliminate the development of resistant mutants.]

We appreciate the reviewer's question regarding the effectiveness of the method. Ultimately, we believe that this question is best answered in the clinic. We recently published clinical results in *The Lancet Infectious Diseases* in which LBP-EC01 with concomitant trimethoprim/sulfamethoxazole (TMP/SMX) achieved a 100% clinical cure and 88% microbiological cure rate (14 of 16 patients) in patients with urinary tract infections caused by drug-resistant *E. coli*, despite 69% of these patients' baseline isolates being TMP/SMX resistant. This year we expect to complete enrollment for the double-blind, placebo-controlled part 2 of this Phase 2 trial, which will allow us to answer this question directly. We intend to publish these data.

In the meantime, we must rely on *in vitro* assessments to determine cocktail effectiveness against a panel of bacterial strains (as *in vivo* models are limited in the number of strains that can be assessed). We believe the most relevant metric to assess the effectiveness of the selected cocktail is the CFU reduction assay as it has the broadest dynamic range. In this assay, LBP-EC01 reduced bacterial titer by 4 log₁₀ in 94% of strains at the 4-hour timepoint and 75% of strains at the 24-hour timepoint. Although there remains room for improvement, this phage cocktail has the broadest host range and most durable suppression of bacterial growth *in vitro* of which we are aware. Groups developing phage therapy do not typically perform the CFU reduction assay on a panel of several hundred bacterial strains due to its time- and consumable-intensive nature. In most cases, groups will publish their total host range as assessed by an optical density assay and/or plaquing host range assay. We note that we have identified cocktails that have similar total host ranges but have marked differences in abilities to reduce bacterial titer and prevent bacterial regrowth in the CFU reduction assays. We find that this is often the case when a common mutation that confers resistance to one phage also confers "cross resistance" to other phages in the cocktail, a situation that our cocktail selection algorithms attempt to avoid and which we believe is related to shared aspects of the infection mechanism (i.e., adsorption to receptor or downstream infection).

Finally, we have focused this paper on the automated methods used to collect phage characterization data rather than the cocktail recommendation algorithms, due to their proprietary nature. However, we agree with the reviewer that phages in the cocktail should leverage different receptors to prevent resistance development, and we note that our selection

metric optimizes for multi-phase coverage of each strain in the panel in a manner that prefers greater genetic diversity of the phages that contribute to that coverage.

3. What is the rate of (detection) error? Is it 100% accurate? As this article is all about the methods, it is important to discuss the inaccuracies in the detection.

For the computer vision detection of colonies, the rate of error was calculated based on a manual assessment of colony counts performed across 3 independent analysts. 768 colonies were assessed and 0.5% of colony counts/titers were determined to be incorrectly calculated. We defined an incorrect calculation as a bacterial titer that was greater than one \log_{10} difference from refereed counts provided by the third operator that was used to investigate differences observed between manual counts and the computer vision method. For example, if the manual count showed a titer 7.5×10^8 , and the computer vision method showed 5×10^6 , this would reflect a greater than one \log_{10} difference, and the calculation would be marked as incorrectly calculated. These values are provided in the section entitled "Automated colony enumeration via computer vision". We are unsure if there are other rates of error that are requested by the reviewer, but we invite any such requests.

Minor:

1. Line no. 108: Are the isolates from urine samples? Need clarity.

We have updated this line to include a reference to the methods section. In this section we have included the following clarifying sentences. "All isolates originated from female urinary tracts. In most cases, these isolates were from urine. A minority of isolates were collected from catheter-obtained urine (n=17) or were labeled by the providing vendor as 'Urinary Bladder' (n=3)."

2. Line no. 109: Check: 98% of strains or isolates?

We have updated this line to specify that these are isolates. Each isolate was not independently verified as being a unique strain; however, across the entire panel, isolates were selected to represent a broad genetic space.

3. Line no. 150: What are those phages belonging to?

If the reviewer is referring to taxonomic classifications, we did not perform this type of assessment on every bacteriophage present in the bank. If this characterization is not what the reviewer is requesting, we invite further clarification and will include the information in the manuscript if we have it.

4. Line no. 169: To be more accurate, the growth media is related to bacteria which eventually relates to phage propagation.

We thank the reviewer for pointing out the inaccuracy and have updated this line to specify that phages are assessed against bacteria grown in different media.

5. Line no. 246,247: LBP-EC01 is one of the 28 cocktails or different.

LBP-EC01 was one of the 28 cocktails that was assessed and was selected as it included the smallest number of phages while maintaining maximum efficacy as assessed by the CFU reduction assay.

6. Line no. 288: It should be '38 stains'.
"Isolates" has been corrected to "strains" in the revised manuscript.

7. Line no. 303-327: Correct the typo in degree sign.
These types have been corrected in the revised manuscript.

Reviewer #2 (Remarks to the Author):

This manuscript contains some impressive results that move phage cocktail therapy toward clinical trials. The problem is that large sections of the paper currently read like a press release. The bulk of the potential contribution to the field is in the data itself on phage-bacteria interactions, which is not released, and the 'proprietary cocktail prediction model', which is not described and can't be replicated.

We think with some edits this could be a very nice paper. It is a powerful story that stretches all the way from the nuts and bolts of engineering a scalable data collection solution for the problem, using that data to solve the main barrier to application, and then applying it to a problem of clinical relevance. The edits that would strengthen the paper is describing how the data is acquired *and* how that data is used to add value.

We appreciated seeing the robotic methods you arrived at for gathering reproducible data on phage infection at high throughput. This is the most technically detailed section of the paper, which we appreciate. With slightly more information (we make some suggestions below), these methods could be replicated and would be very useful for other phage biologists.

We thank the reviewer for their helpful feedback and agree with the commentary provided here. Please see the below in line comments on the changes we have made to improve the manuscript. Although we are restricted in our ability to share proprietary aspects of the platform, we agree with the reviewer that additional code, methods, and data are needed to improve the reproducibility of the manuscript. We are now providing the fundamental pieces to enable other groups to replicate the generalized process of using robotics and bioinformatics to capture data characterizing phage efficacy. This includes the release of our automation code to facilitate other researchers, as recommended by the reviewer.

Regarding the cocktail prediction model, we respectfully note that this model incorporates proprietary design elements developed as part of an ongoing therapeutic program and cannot be shared in full. However, we have added more detail in the in-line comments below to respond to the reviewer's recommendations. We hope this provides sufficient transparency into the model's role in cocktail development while preserving necessary confidentiality. We offer the analogy of publication of new foundational large language models, in which it is widely accepted that authors will publish model architecture without sharing proprietary parameter weightings.

We appreciate the feedback regarding the tone of the paper and have updated several areas of the manuscript to remove language that contributes to this sentiment. We used an LLM to perform sentiment analysis and highlight regions of the paper that should be modified. We have revised the abstract as well as several other locations throughout the paper, which are

highlighted in the attached manuscript. Although we endeavored to be comprehensive in these revisions, we are open to further revisions if we have missed any such instances.

Suggestions on edits to robotic methods for gathering phage infection data

Would the authors be willing to release their automation code? They are presumably saved as a Venus file. The code contains a lot of details that would be valuable for using this technique again, i.e. liquid classes that are used for different reagents, whether liquid handling steps have been optimized to avoid traversing over other plates to avoid contamination, etc.

Figure 1 contains renders of their instrument. It would be more helpful to have a photograph, a deck layout image, and/or a visual depiction of the method itself (see an example in figure 2a of <https://www.embopress.org/doi/full/10.15252/msb.20209942>)

Could the authors provide more details and the automated preparation of the plaque forming assays? At the moment, it is not clear how the PFU assays are prepared - as described later in the manuscript one would expect a double agar overlay method here to establish a bacterial lawn on some of the plates. For the PFUs, are the phages separated from the bacteria before having samples plated?

We appreciate the reviewer's interest in the technical aspects of our robotic workflows and fully agree that the reproducibility of high-throughput experimental methods is essential for broader adoption in the field. In response to this comment, we will make the INSTINCT V automation code files used for the optical density assay publicly available. This method includes the detailed parameters such as liquid classes for various reagents, deck layout, pipetting behaviors, and safeguards implemented to minimize cross-contamination.

We believe that sharing this level of detail represents a meaningful and actionable contribution to the field, particularly for researchers aiming to implement or adapt similar high-throughput robotic systems. The code will be made public via a persistent online repository at the time of manuscript publication.

Furthermore, we agree with the reviewer's suggestion that including a deck layout image that highlights the placement of labware will be useful for the manuscript. We have included an additional supplementary figure that details this information. We anticipate this addition will assist other researchers in replicating or adapting our setup for similar high-throughput applications and complements the release of the automation code.

Finally, we have added additional details into the methods section titled "Automated Colony and Plaque Enumeration Assay." These additions should clarify the details of how the double agar overlay assay was performed.

Suggestions on edits to methods for using phage data to predict effective phage cocktails:

Currently the extent of detail is this sentence: "We developed a proprietary cocktail prediction model that leveraged our database of phage and bacteria reactions to recommend cocktails for additional testing." Figure 2 shows histograms of cocktail effectiveness, the top cocktails and their effectiveness against UTI strains. We understand that you can't release this algorithm, but it would be valuable to find other ways to discuss this part of the paper with more nuance.

We thank the reviewer for the constructive feedback that comparison to alternative choices would be illustrative of the advantage to the proprietary methods that were employed. Details regarding the genomic elements of our proprietary methods are captured below. We agree that adding additional nuance would be helpful and have included this as part of our response to the following reviewer comment.

Could you show the performance of the proprietary method relative to simple control algorithms for selecting phage cocktails? For example, you should show Figure 2c for the proprietary cocktail selection method vs random vs another simple method (say, regression based). This would help other scientists understand whether they could use a simpler, less effective technique and still have it work, or whether they should expect to need to develop something more sophisticated if they wish to attempt this sort of technique.

How much better is the proprietary method?

How many fewer experiments did it allow them to run?

How iterative/closed loop was it?

How did they handle the explore/exploit tradeoff?

We thank the reviewer for suggesting a comparison to simple control algorithms. We have completed and added an additional analysis to the manuscript that compares alternative cocktail generation methods to the results of the full proprietary methods. The two added simple algorithms are based on the area under the curve for optical density up to 20 h of growth. These two methods do not contain the genomic analyses performed for final cocktail selection, or the proprietary ranking and host range distance metrics employed. Full details are in the revised manuscript, but the added algorithms for comparison are "random choice of 1 to 6 phages combined into a cocktail" and a "ranked choice of 1 to 6 phages with highest host ranges combined into a cocktail". An additional plot, "Extended Data Fig. 9", has been added to the supplemental information to summarize this analysis. A few sentences summarizing the results of this additional analysis have been added to the body of the manuscript. The full proprietary method for cocktail development, including synthetic modifications, leads to a final cocktail with ~97% host range activity, which is noticeably higher than the ~90% host range activity that would be suggested from simple algorithmic wild-type phage selection and their predicted combined efficacy when assembled into cocktails.

We did not directly calculate or extrapolate the reduction in experiment count that the proprietary methods should have allowed, as our target was generally to err on the side of additional experiments to support the preclinical efforts whenever possible. The proprietary methods were closed loop in the sense that we used rarefaction of both cocktail efficacy as well as rarefaction of genetic diversity to assess when continued effort led to diminishing returns. This same rarefaction was used to balance the tradeoffs between exploration and exploitation.

While we won't be able to comment on the relative performance of different cocktail selection methods for therapeutic efficacy in patients, our double-blind, placebo-controlled clinical trial using the selected cocktail will read out later this year (<https://clinicaltrials.gov/study/NCT05488340>) and will be the subject of a subsequent publication. We will compare the predicted host range with the observed host range in the clinic.

These are the fundamental questions that is of a lot of interest to the field broadly. A lot of the potential value of your paper is interpreting it as a specific scientific example of the value of

robotics + data analysis in solving high-combinatorial problems in life science.

The dataset itself, if released, would be valuable: 421 phages to 356 E. coli, which is more than the current state of the art: 96 phages to 403 E. coli from Pasteur recently (data <https://zenodo.org/records/13831957>, and paper <https://www.nature.com/articles/s41564-024-01832-5>). If it's not possible to release the data, perhaps you could use your computational tools on these public datasets versus your private dataset

We thank the reviewer for this suggestion. We are now sharing the liquid host range data, included in the plots of Fig. 2, to improve understanding of the analysis. Shared data sets include both early development screening (100 strains tested against 170 phages) as well as later stage testing against the full 357 strain Clinical Panel (28 phage cocktails). However, due to our active pharmaceutical development program, we cannot share the entire phenotypic or genomic dataset. It would be possible with significant effort to format and condition the data to run our tools against the public dataset in the Pasteur paper, however, because the assays used as inputs for the prediction methods are distinct, we feel that an additional complete analysis against a novel dataset in the Pasteur publication exceeds the scope of this manuscript. We feel the Pasteur data is complementary to the methods and results presented in our present manuscript. We would be happy to participate in a follow up manuscript in collaboration with the Pasteur institute, should the editors find merit in comparing results between tools, as well as potentially a larger meta-analysis of additional data sets across the growing community of phage and bacterial genomes researchers.

Other comments by line:

Fig 3 d and e: What do the different colours mean in this figure?

Additional text was added to the figure legend to indicate that the different colors indicate uniquely identified colonies and plaques.

Line 47: Reference 5 estimates annual deaths attributable to AMR to be 1.91 million (1.56–2.26) and 8.22 million (6.85–9.65) deaths associated with AMR in 2050. Because of the context of the sentence which uses the language “an estimated 1.27 million deaths were attributed to bacterial antimicrobial resistance (AMR) [2], and this number is estimated to be as high as 10 million by 2050”, I think that either the wording should be changed to show that the 10 million (from ref 4) is not referring to attributable deaths or the 1.91 million number should be given.

We thank the reviewer for pointing out this error and have updated to the following language: “In 2019 an estimated 1.27 million deaths were attributed to bacterial antimicrobial resistance (AMR) [2], and deaths associated with AMR are estimated to be reach 8.22 (6.85–9.65) million by 2050 [4, 5].”

Line 70: “their favorable safety profile” - should have a reference

We have added a citation to support this sentence from a publication created by the Antibacterial Resistance Leadership Group.

Line 138, 147, 347: Shaking speed of the plates is important due to the effect on aeration (Shaer Tamar and Kishony 2022) (see additionally ref 30-38 of this paper) - shaking speed should be mentioned in the main text and orbital radius should be given in methods.

We agree that this information is important to include and have added the shaking speed used for the phage discovery process into the main text. Additionally, we have added information into the methods about the orbital shaker amplitude, which is 2 mm.

In line 347, why was 250rpm chosen for *E. coli*? What data informed this decision?

To address this question, the following text was added to the manuscript at line 389:

“This shaking speed was selected based on initial growth optimization experiments for a random selection of *E. coli* strains normalized to a starting optical density of 0.02, in which various speeds between 100 and 800 rpm were studied. Of these, 250 rpm was selected because it optimized for maximal area under the bacterial growth curve while minimizing the speed. Minimizing speed was an optimization objective, as it reduces both the risk of cross well contamination within the 384-well plates and the long-term stress on the automated shaking incubators.”

Line 169 and 172: What conditions were tested? What components of the media were varied? How did this affect the growth of the different bacteria and phages?

Some have observed phages showing limited efficacy in human serum - was this tested?

In these lines, we note that multiple experimental conditions were tested. Our intent in this section was to highlight that the conditions of our *in vitro* assay may not translate to conditions present in acute urinary tract infections. A benefit of using automated systems is the ability to apply scale to test many different conditions and mitigate this risk. We have added a line to this section to better match our intent.

Key conditions that we tested included: 1) varying bacterial titer and phage titer inputs, 2) various matrices such as lysogeny broth, synthetic urine, and pooled human urine, 3) presence of additives such as $MgCl_2$ and $CaCl_2$, and 4) bacterial growth state at the time of phage exposure. As expected, higher multiplicity of infection (MOI) led to improved phage efficacy, and bacteria grew more slowly and to a lower final optical density in urine compared to lysogeny broth. We did not observe large changes in phage efficacy when comparing urine to lysogeny broth. We intend to perform a comprehensive analysis of varying conditions and correlate them to the patient bacterial reduction data that we are capturing as part of our ongoing Phase 2 trial. *E. coli* isolates recovered from patient urine throughout the treatment time course will be exposed to LBP-EC01 under different experimental conditions, and we will determine the conditions that best correlate to observed *E. coli* level changes in patients. We believe these data will be best suited for follow-on publication after the Phase 2 data collection is completed.

Regarding testing in human serum, we recently published the resulting phage levels after IV dosing in part 1 of a Phase 2 trial¹. In patients with an IV dose of 6 mL of 1×10^{11} PFU/mL drug product, phage levels in blood samples peaked between 1×10^6 and 1×10^7 PFU/mL. Assuming the average patient has a blood volume of 5000 mL, the maximum theoretical blood concentration would be $\sim 1.2 \times 10^8$ PFU/mL, indicating that we are observing 1-10% of the maximum theoretical pharmacokinetic signal. The reduction between our observation and the maximum possible pharmacokinetic signal is likely due to rapid sequestration of phages to the spleen and liver, as previously reported in primates:

<https://pmc.ncbi.nlm.nih.gov/articles/PMC10805017/>.

Line 217: Figure 4 doesn't have any PFUs in the axis titles but the text here says that PFUs are represented in the data - this is confusing

Thank you for catching this error. We have removed the reference to PFU in this sentence.

Line 300: If including this line, then state what method was used to come to this conclusion and ideally show the data

We have updated this section to include more detailed language about the actual testing that was completed and the thresholds that were used to determine whether resistance to the cocktail had developed. Additionally, we have added clarification to ensure that the reader understands that we were searching for genetic-based resistance mechanisms, as we cannot rule the possibility of transient resistance to LBP-EC01 *in situ*. The following text was added to the manuscript:

“Finally, we assessed sensitivity of all 166 *E. coli* isolates to LBP-EC01 by plaquing and optical density time course phage efficacy assays. For each patient, we compared the genome assemblies of urine-derived *E. coli* isolates collected pre-treatment and across a 34-day period after initial treatment. Because some patients harbored multiple *E. coli* strains, we grouped isolates by sequencing type within each patient. Resistance to LBP-EC01 was defined as a loss of phage efficacy within the same sequencing type over time—either a >1-log reduction in plaquing efficiency or a shift from susceptible to resistant classification in the optical density assay (AUC ratio threshold = 0.65). Using this approach, we did not observe the development of genetic resistance to LBP-EC01 in any patient enrolled in Part 1 of the study”.

Line 330: I could not find any information online about what Modules A,B,C, and D of the Hamilton Vantage 2.0 system are

We have included an additional sentence in the methods section that describes the different modules of the system, as Hamilton may not provide the module definitions on their website. We assume the inclusion of the modules may help others examine potential similar systems.

“Module A refers to the pipetting deck, while Module B is the logistics cabinet that supports the pipetting deck. Module D is the rear cabinet containing the incubators and plate readers. Positioned between Modules A/B and D, Module C is the track gripper or robotic arm that facilitates plate movement across the system.”

Reviewer #2 (Remarks on code availability):

Some methods are described, but the code isn't linked in the methods unless we're missing something?

Please see the above section detailing the automation code we are releasing to improve the reproducibility of the methods.

Reviewer #3 (Remarks to the Author):

Reviewer #3 (Remarks on code availability):

Not available

Citation:

- 1) Kim, Paul, et al. "Safety, pharmacokinetics, and pharmacodynamics of LBP-EC01, a CRISPR-Cas3-enhanced bacteriophage cocktail, in uncomplicated urinary tract infections due to Escherichia coli (ELIMINATE): the randomised, open-label, first part of a two-part phase 2 trial." *The Lancet Infectious Diseases* 24.12 (2024): 1319-1332.

Response to Reviewers Round 2

High-throughput methods leveraging robotics and computer vision for the development and assessment of therapeutic phage cocktails

REVIEWER COMMENTS

Reviewer #2 (Remarks to the Author):

These edits have greatly improved the paper and we appreciate the detailed answers to all of our comments.

We are grateful to the reviewers for facilitating the paper improvement, and we were happy to provide details to the helpful questions and feedback.

Thank you for going to the effort of using OCR to get the code out of the INSTINCTV software for others to read and for sharing the GIFs of the simulations! We hope this makes your methods more widely accessible.

We were happy to provide this update and have included this code in the publicly facing repository.

Cocktail selection analysis: The addition of Extended Data Figure 9, comparing your methods to random and top-k selection algorithms, is very helpful. Given its significance in illustrating how your proprietary active learning algorithm enhanced exploration of the phage sequence space, this figure might warrant consideration for inclusion in the main text.

We agree with the recommendation and have added this figure into the main text.

Citation Correction: On a second read, we followed the citation chain for “The frequency and recurrence of UTIs are a significant health burden and are associated with an estimated \$1.6 billion annual cost [12]”. The only reference in 12 that seems to be for the \$1.6 billion number is to a paper entitled “Risk factors for second urinary tract infection among college women” (from the line: “The estimated annual cost of community-acquired UTI is approximately \$1.6 billion.[15]”. This paper is about how the sexual behaviours of college women from 1992 to 1994 affected their probabilities of getting a repeat UTI - very interesting but there is no mention of the \$1.6 billion number! We request this citation be removed before publication. Rather than attempting to replace it with another economic estimate, we suggest the human health burden data already presented is sufficient to make your case.

We're grateful to the reviewer for catching this error and agree with their recommendation to remove this citation. We have updated the manuscript accordingly.

We wish your clinical trial success and hope your future work leads to greatly improved patient outcomes.

Reviewer #2 (Remarks on code availability):

The authors share Hamilton automation code for the three assays used in the paper. To help others who try to replicate or use the code, they share videos of the simulations of the experiments when run on their hardware and software. Furthermore, for those with different hardware and software setups they share the raw text of the automation code for each assay (outside of the hardware manufacturer's proprietary file format).

We note that all of these provided files and simulations have been added to the publicly facing repository.